# CNNM proteins selectively bind to the TRPM7 channel to stimulate divalent cation entry into cells

Zhiyong Bai[1], Jianlin Feng[2], Gijs A. C. Franken[3], Namariq Al'Saadi[1,4], Na Cai[1], Albert S. Yu[2], Liping Lou[1], Yuko Komiya[1], Joost G. J. Hoenderop[3], Jeroen H. F. de Baaij[3], Lixia Yue[2], Loren W. Runnels[1]*

**1** Rutgers-Robert Wood Johnson Medical School, Piscataway, New Jersey, United States of America, **2** UCONN Health Center, Farmington, New Mexico, United States of America, **3** Radboud University Medical Center, Nijmegen, the Netherlands, **4** University of Misan, Amarah, Iraq

* runnellw@rwjms.rutgers.edu

## Abstract

Magnesium is essential for cellular life, but how it is homeostatically controlled still remains poorly understood. Here, we report that members of CNNM family, which have been controversially implicated in both cellular $Mg^{2+}$ influx and efflux, selectively bind to the TRPM7 channel to stimulate divalent cation entry into cells. Coexpression of CNNMs with the channel markedly increased uptake of divalent cations, which is prevented by an inactivating mutation to the channel's pore. Knockout (KO) of *TRPM7* in cells or application of the TRPM7 channel inhibitor NS8593 also interfered with CNNM-stimulated divalent cation uptake. Conversely, KO of *CNNM3* and *CNNM4* in HEK-293 cells significantly reduced TRPM7-mediated divalent cation entry, without affecting TRPM7 protein expression or its cell surface levels. Furthermore, we found that cellular overexpression of phosphatases of regenerating liver (PRLs), known CNNMs binding partners, stimulated TRPM7-dependent divalent cation entry and that CNNMs were required for this activity. Whole-cell electrophysiological recordings demonstrated that deletion of *CNNM3* and *CNNM4* from HEK-293 cells interfered with heterologously expressed and native TRPM7 channel function. We conclude that CNNMs employ the TRPM7 channel to mediate divalent cation influx and that CNNMs also possess separate TRPM7-independent $Mg^{2+}$ efflux activities that contribute to CNNMs' control of cellular $Mg^{2+}$ homeostasis.

## Introduction

More and more it is recognized that cells are molecular machines, with intricate components working seamlessly together to regulate its multitude of activities. Magnesium and other divalent cations are among the cells most essential molecules. With an intracellular concentration of 14 to 20 mM, $Mg^{2+}$ is the second most abundant cation after $K^+$ [1,2]. Bound to ATP and other macromolecules within the cells, it has been estimated that at least 600 enzymatic reactions are directly or indirectly regulated by $Mg^{2+}$ [3]. Consequently, dysfunction of

**Data Availability Statement:** All relevant data are within the paper and its Supporting information Files. Additional supplementary data for the mass spectrometry experiments have been deposited

into the ProteomeXchange database (Accession Number: PXD026635) and are publicly available.

**Funding:** Research was supported by a grant from National Heart, Lung, and Blood Institute, National Institutes of Health, to L.Y. and L.W.R. (R01HL147350). The funders had no role in study design, data collection and analysis, decision to publish, or preparation of the manuscript.

**Competing interests:** The authors have declared that no competing interests exist.

**Abbreviations:** ACDP, ancient conserved domain protein; BioID, biotin identification; CBS, cystathionine beta-synthase; DMEM, Dulbecco's Modified Eagle Medium; FDR, false discovery rate; HBSS, Hanks' balanced salt solution; IMEM, improved minimal essential media; KO, knockout; NASH, nonalcoholic steatohepatitis; OK, opossum kidney proximal tubule; PRL, phosphatase of regenerating liver; PRL-2, phosphatase of regenerating liver 2; WT, wild-type.

magnesium regulation has been linked to many human diseases including cancer, diabetes, epilepsy, muscle cramps, T-cell immunodeficiency, and Parkinson disease [3]. Unlike $Ca^{2+}$, whose cytosolic concentration can vary 10- to 100-fold, the free concentration of $Mg^{2+}$ in cells is comparably steady, hovering between 0.4 and 1.2 mM in concentration. The mechanisms by which intracellular $Mg^{2+}$ homeostasis is maintained is complex, relying on the activities of transporters and ion channels whose regulation and coordination is still poorly understood.

The ubiquitously expressed TRPM7 was the first identified channel with its own kinase domain and is also notable for its permeability to $Mg^{2+}$ [4]. Loss-of-function analysis have revealed that TRPM7 is essential for $Mg^{2+}$ uptake in multiple cell lines [5–9] and significantly contributes to embryonic development and organogenesis [10–12]. As a divalent-selective channel, TRPM7 has also been found required for cellular and bodily uptake of $Zn^{2+}$ [13,14]. Influx of $Ca^{2+}$ through the channel has been also been found necessary for cell migration [15] and invadosome formation [16]. Dysregulation of TRPM7 contributes to the pathophysiology of a large variety of disorders, especially cancer [17,18], cardiovascular defects [19,20], and neuronal injury caused by stroke and traumatic brain injury [21–23]. Despite its medical importance, regulation of the channel in vivo is poorly understood, and few interacting proteins have been identified.

Members of the CNNM family have also been linked to the regulation of $Mg^{2+}$ homeostasis. In humans and mouse, the CNNM family, previously referred to as ancient conserved domain proteins (ACDPs), has 4 members, CNNM1, CNNM2, CNNM3, and CNNM4, which are differentially expressed throughout the body [24,25]. CNNMs are transmembrane membrane proteins that contain a Bateman module composed of 2 cystathionine beta-synthase (CBS) motifs [26–28]. The Bateman module of CNNMs bind to $Mg^{2+}$-ATP [28], which has been demonstrated to cause significant structural changes in the Bateman module [26], affecting the orientation of the domain with the CNNM transmembrane region [29]. CNNMs have also been found to be regulated by members of the phosphatases of regenerating liver (PRLs) family, which also bind to the Bateman module [30–32]. Recently, ARL15, which can also bind to CNNMs, was also found to influence CNNM function by affecting its glycosylation [33].

Like TRPM7, CNNMs have been associated with important physiological and pathological activities. CNNM2 is involved in the regulation of sperm motility and blood pressure regulation [34,35]. The isoform is also highly expressed in brain, with mutations in *CNNM2* associated with schizophrenia [36,37], brain malformations, and epilepsy [34,38]. CNNM3 and CNNM4 have been associated with oncogenesis through their physical association with PRLs [30,31]. More recently, up-regulation of CNNM4 has been shown to contribute to the development of nonalcoholic steatohepatitis (NASH) [39]. In addition, CNNMs have been recognized as having important functions in polarized epithelial cells in the kidney and intestine. Mutations in *CNNM2* cause hypomagnesemia [40]. CNNM4 has been demonstrated to be required for basolateral mediated $Mg^{2+}$ transport in the intestine [41].

Despite CNNM proteins' obvious clinical importance, identifying the specific functional activity of these proteins has been the subject of intense debate [42,43]. Evidence that CNNMs mediate $Mg^{2+}$ efflux is particularly strong for CNNM2 and CNNM4, which have robust $Mg^{2+}$ efflux activities compared to CNNM1 and CNNM3 [28]. Consistent with these experiments, CNNM2 has been found to be a direct transporter that extrudes $Mg^{2+}$ ions from the basolateral side of polarized epithelial cells [40,41]. Yet, other experiments have found evidence that $Mg^{2+}$ uptake can also be stimulated by CNNM2 [38], while still another study alternatively concluded that CNNM2 is not directly involved in membrane transport but instead acts as either intracellular $Mg^{2+}$ sensor or as $Mg^{2+}$ homeostatic mediator of other not-yet-identified transcellular transporters that alternatively regulate divalent cation influx and efflux processes [44]. In addition to CNNM2, CNNM3, when bound to the protein tyrosine phosphatase of

regenerating liver 2 (PRL-2), has also been reported to mediate $Mg^{2+}$ uptake [31,32]. Thus, CNNMs have been implicated in both cellular $Mg^{2+}$ influx and efflux, and despite their physiological and pathological importance, their specific functional activities and how they are controlled remain unclear.

Here, we seek to resolve the controversy in the field by providing evidence that CNNMs have the capacity to influence 2 opposite sides of one essential activity in cells—$Mg^{2+}$ transport. Our experiments demonstrate that CNNMs employ the TRPM7 channel to accomplish divalent cation influx and that in the absence of the channel, CNNM2 and CNNM4, in particular, can potently lower intracellular $Mg^{2+}$ levels. The identification of CNNMs as a regulatory factor for the TRPM7 channel opens up new opportunities for understanding the channel's in vivo regulation as well as for probing the new found functional relationship between CNNMs and TRPM7.

## Results

### CNNMs interact with TRPM7 and regulate its activity

To identify interacting proteins of the TRPM7 ion channel, we conducted mass spectrometry analysis of FLAG-tagged mouse TRPM7 from a tetracycline-inducible HEK-293 cell line (LTRPC7 cells herein referred to as 293-TRPM7 cells) maintained in the absence of tetracycline to keep channel expression low [4,45]. HEK-293T cells were used as a negative control. The mass spectrometry experiment uncovered CNNM proteins as potential TRPM7-interacting proteins, as well as PTP4A1 (PRL-1) and ARL15, which are known CNNM binding proteins (S1 Table) [30,31,33]. Since CNNM proteins and TRPM7 have both been associated with the regulation of $Mg^{2+}$ homeostasis, we were motivated to investigate the relationship between these proteins. To evaluate the interaction between CNNMs and TRPM7 uncovered by our mass spectrometry experiments, we first validated the ability of the proteins to interact. Immunoprecipitation of HA-tagged TRPM7 (HA-TRPM7) that was transiently transfected into HEK-293T cells coimmunoprecipitated all transiently expressed FLAG-tagged CNNM isoforms (Fig 1A). TRPM6, a close homologue with channel and kinase domains, also interacts with CNNM proteins (S1 Fig). Interestingly, our experiments revealed that overexpression of CNNM2 and CNNM4 with TRPM7 significantly decreased the channel's protein expression (Fig 1A), which reduced that amount of CNNM2 or CNNM4 that copurified with TRPM7. Overexpression of CNNM2 and CNNM4 also decreased the expression of other proteins, indicating that this effect is nonspecific. For example, coexpression of CNNM2 or CNNM4 with GFP also reduced GFP protein expression (S2 Fig). Since CNNM2 and CNNM4 have been reported to have robust $Mg^{2+}$ export activities [28], it is possible that overexpression of these protein interfered with protein translation because of their ability to lower of cellular $Mg^{2+}$ levels. Using antibodies specific to CNNM3 and CNNM4, 2 isoforms that were identified in the mass spectrometry screening, we found that immunoprecipitation of overexpressed HA-TRPM7 coimmunoprecipitated native CNNM3 and CNNM4 proteins (Fig 1B), indicating that when the channel is overexpressed in HEK-293 cells, it forms complexes with the endogenous CNNM3 and CNNM4 proteins. The interaction of TRPM7 with CNNMs is specific, as another TRP channel family member, TRPM2, did not interact with the native CNNM3 and CNNM4 proteins (Fig 1B). We also found that when overexpressed, HA-TRPM7 has the capacity to bind native CNNM proteins in other cell lines (Hela, opossum kidney proximal tubule (OK), RPTEC/TERT1, and HAP1) and that endogenous TRPM7 interacts with native CNNM4 in ZR-75-1 breast cancer cells (S3 Fig). Thus, the interaction between TRPM7 and CNNMs can be observed in multiple cell systems. Both TRPM7 and CNNM proteins have been reported to localize and function at the plasma membrane [28,46,47]. Consistent with

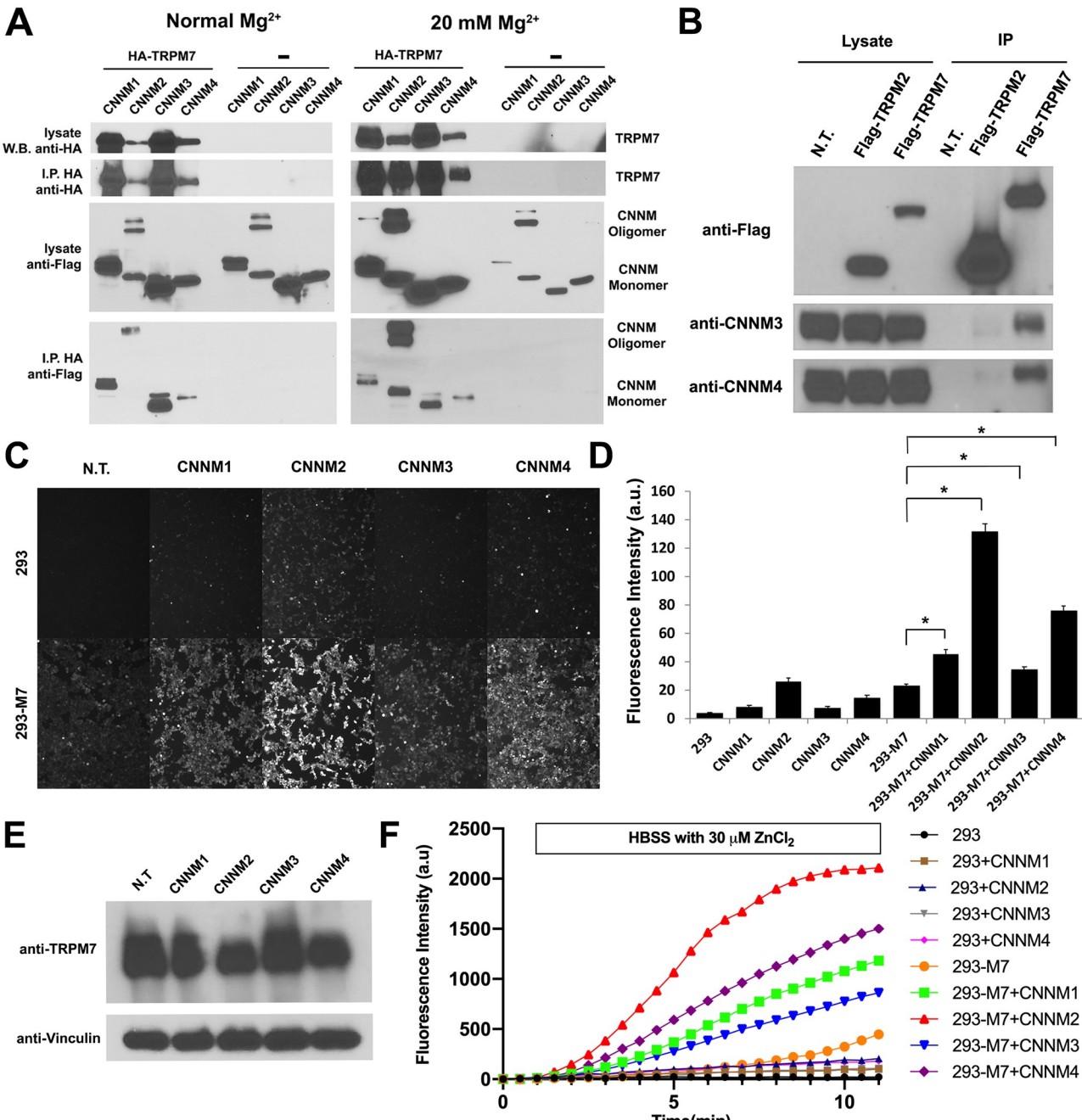

**Fig 1. CNNMs interact with TRPM7 and regulate its activity. (A)** HA-TRPM7 was coexpressed with FLAG-tagged CNNM1-4 in HEK-293T cells, and the channel was immunoprecipitated with HA-agarose. CNNM1 and CNNM3 strongly interacted with TRPM7. CNNM2 and CNNM4 also interacted with TRPM7, but lowered TRPM7 expression nonspecifically. CNNMs were not coimmunoprecipitated from HEK-293T cells N.T. with HA-TRPM7. Supplementation of the growth medium with 20 mM $MgCl_2$ boosted TRPM7 expression, revealing the channel's capacity to interact with CNNM2 and CNNM4. **(B)** FLAG-tagged TRPM7 robustly coimmunoprecipitated native CNNM3 and CNNM4, whereas FLAG-TRPM2 did not. **(C)** A Zinc influx assay using the FluoZin-3 $Zn^{2+}$ indicator was used to monitor TRPM7 function in intact 293-TRPM7 cells (293-M7 cells), which express FLAG-tagged TRPM7 upon tetracycline treatment. Coexpression of CNNMs with TRPM7 increased intracellular free $Zn^{2+}$ more than expression of TRPM7 alone. The T-REx-293 cell line (293), which expresses the Tet Repressor protein, was used as the negative control. Shown are images taken at a time point between 5 and 10 minutes after application of 30 μM $ZnCl_2$. White scale bar = 100 μM. **(D)** Quantification of the results from (C). A total of 100 cells were randomly selected for quantification. $n$ = 100. * indicates a $p$-value of less than 0.05. **(E)** Western blot showing expression of TRPM7 in the sample from (C). **(F)** Separate time course measurements were also acquired to further demonstrate that cells coexpressing CNNMs with TRPM7 have a higher rate of $Zn^{2+}$ influx compared to 293-TRPM7 cells expressing TRPM7 or CNNMs alone. HBSS media was replaced with HBSS containing

30 μM ZnCl$_2$ for the period indicated. The fluorescence intensity of the cells (mean of 50 cells) were quantified for each time point. Unprocessed images of blots are shown in S1 Raw Images. The underlying data for this figure can be found in S1 Data. HBSS, Hanks' balanced salt solution; N.T., not transfected.

published data, we observed that HA-TRPM7 and FLAG-tagged CNNM3 can be found colocalized in HEK-293T cells at the cell border (S4 Fig), although considerable TRPM7 staining can also be found intracellularly. To further assess these proteins' association, we employed proximity-dependent biotin identification (BioID), a recently developed method that allows the identification of proteins in the close vicinity (10 to 30 nm) of a protein of interest in living cells [48]. We employed the optimized Escherichia *coli* BirA biotin ligase (TurboID) to engineer a fusion protein between CNNM3 and TurboID (CNNM3-FLAG-TurboID) [49]. CNNM3-TurboID efficiently labeled TRPM7 with biotin compared to a negative control EYFP-FLAG-TurboID, adding evidence that CNNM3 is vicinal to TRPM7 when the 2 proteins are coexpressed in HEK-293 cells (S5 Fig).

We next sought to determine the functional significance of the interaction of CNNM proteins with TRPM7. Since CNNM proteins have been reported to influence both Mg$^{2+}$ influx and efflux, we took advantage of the fact that TRPM7 is divalent selective, with highest permeability to Zn$^{2+}$, to assay TRPM7-mediated divalent cation influx [50]. Application of 30 μM ZnCl$_2$ to the extracellular solution of HEK-293 cells stably overexpressing TRPM7 (293-TRPM7) readily produced a detectable increase in intracellular free Zn$^{2+}$, as monitored by the zinc indicator FluoZin-3 (Fig 1C–1F). We observed significantly higher Zn$^{2+}$ influx in cells coexpressing CNNMs with TRPM7 than in cells that express TRPM7 alone (Fig 1C–1F). Coexpression of TRPM7 with CNNM2 and CNNM4 produced the greatest stimulation of TRPM7-dependent Zn$^{2+}$ uptake. By transfecting modest concentrations of CNNM cDNA, we were able to minimize CNNM-mediated interference with TRPM7 protein expression (Fig 1E). Interestingly, we observed that overexpression of either CNNM2 or CNNM4 in wild-type (WT) HEK-293 cells produced a small but detectable increase in Zn$^{2+}$ uptake (Fig 1C–1F). To determine whether native TRPM7 was required for this CNNM-mediated Zn$^{2+}$ influx activity, we expressed CNNM2 in HEK-293T TRPM7 knockout (KO) cells (293TΔM7) and measured Zn$^{2+}$ uptake [14]. The absence of TRPM7 in HEK-293T cells prevented a CNNM2-dependent increase in intracellular Zn$^{2+}$ (S6A–S6D Fig), indicating that the endogenous TRPM7 channel is the major contributor of CNNM2-mediated Zn$^{2+}$ influx. To confirm these results, we tested whether CNNM2 stimulated Zn$^{2+}$ influx in another cell line in which TRPM7 had been knocked out (HAP1 TRPM7 KO cells) [6]. Similarly, KO of TRPM7 in HAP1 cells blocked a CNNM2-dependent increase in intracellular Zn$^{2+}$ (S7 Fig), indicating CNNMs when overexpressed can stimulate native TRPM7 channel function in different cell types. To further investigate the requirement of the TRPM7 channel for CNNM stimulation of Zn$^{2+}$ uptake, we tested whether coexpression of the TRPM7-E1047K channel-inactive pore mutant with CNNM2 affected Zn$^{2+}$ influx [51]. For these experiments, we used HA-tagged TRPM7 (WT) and TRPM7-E1047K HEK-293 expressing cells made using the Flp-In system to allow isogenic tetracycline-inducible protein expression [47,51]. Coexpresssion of CNNM2 with WT TRPM7, but not TRPM7-E1047K, produced an increase in Zn$^{2+}$ uptake, indicating the TRPM7's pore is required for CNNM2 stimulation of TRPM7-dependent Zn$^{2+}$ influx (S6E–S6H Fig). To further confirm the contribution of the channel to CNNM-mediated divalent cation influx, we employed the TRPM7 channel inhibitor NS8593 [52]; our results show that CNNM-mediated TRPM7-dependent Zn$^{2+}$ influx can also be blocked by this compound (S6D Fig).

TRPM7 is blocked intracellularly by Mg$^{2+}$ and Mg-ATP [4]. This prompted us to investigate whether CNNM stimulation of TRPM7-dependent Zn$^{2+}$ influx could be explained by the

lowering of intracellular $Mg^{2+}$ due to CNNM's $Mg^{2+}$ export activity. Both expression of CNNM4 as well as the $Mg^{2+}$ exporter SLC41A1 [53] modestly suppressed the increase in intracellular $Mg^{2+}$ levels caused by overexpression of TRPM7 in 293-TRPM7 cells (S8A–S8C Fig). However, only CNNM4 robustly stimulated $Zn^{2+}$ influx through the channel (S8A, S8D and S8G Fig), indicating that activation of TRPM7 channel function is specific to CNNMs.

## Native CNNMs influence TRPM7 function in intact cells

Overexpression of CNNMs in HEK-293 cells profoundly stimulated TRPM7 channel function (Fig 1C–1F). We next asked whether endogenous CNNM3 and CNNM4, the most abundant CNNM isoforms found interacting with TRPM7 from our mass spectrometry experiment, affect TRPM7-dependent $Zn^{2+}$ influx in 293-TRPM7 cells. CRISPR/Cas-9 was employed to KO *CNNM3*, *CNNM4*, and both isoforms together from 293-TRPM7 cells (293-M7-ΔCNNM3, 293-M7-ΔCNNM4, and 293-M7-ΔCNNM3/4). Two independent clones were created for each line and validated for loss of CNNM isoforms by SDS-PAGE and western blotting (Fig 2D) as well as by genomic sequencing (see Methods). Unlike TRPM7 KO HAP1 and DT40 cells [6,8], 293-M7-ΔCNNM3, 293-M7-ΔCNNM4, and 293-M7-ΔCNNM3/4 cells do not require $Mg^{2+}$ supplementation to sustain cell proliferation. In addition, KO of CNNMs did not affect TRPM7 protein levels (Fig 2D), nor surface expression of the channel (Fig 2E), as assessed by cell surface biotinylation experiments. We found that KO of *CNNM3* from 293-TRPM7 reduced $Zn^{2+}$ influx roughly by half, compared to the 293-TRPM7 parental cell line, whereas KO of *CNNM4* caused more of a substantial decrease in TRPM7-dependent $Zn^{2+}$ influx (Fig 2A and 2B, S9 Fig). KO of both CNNM3 and CNNM4 together did not further decrease TRPM7-dependent $Zn^{2+}$ influx below that observed in 293-M7-ΔCNNM4 cells (Fig 2A and 2B, S9 Fig). This result indicates that while CNNM3 contributes to TRPM7-mediated $Zn^{2+}$ influx, native CNNM4 appears to be the more potent regulator of the TRPM7 channel in HEK-293 cells. We conducted gain-of-function experiments to investigate whether reexpression CNNM3, CNNM4, CNNM3, and CNNM4 together could recover $Zn^{2+}$ influx (Fig 2A and 2B). Reexpression of CNNM3 in 293-M7-ΔCNNM3 cells was able to stimulate TRPM7-dependent $Zn^{2+}$ influx to levels similar to control 293-M7 cells (Fig 2A–2C). Whereas, reexpression of CNNM4 in 293-M7-ΔCNNM4 cells increased TRPM7-dependent $Zn^{2+}$ influx beyond the level observed for control 293-M7 cells (Fig 2A–2C). Interestingly, reexpression CNNM3 in 293-M7-ΔCNNM3/4 cells did not fully bring TRPM7-dependent $Zn^{2+}$ influx to a level similar to control 293-M7 cells, whereas expression of CNNM4 by itself in 293-M7-ΔCNNM3/4 cells again raised TRPM7-dependent $Zn^{2+}$ influx above the level obtained in 293-M7 cells (Fig 2A–2C). Coexpression of CNNM3 and CNNM4 in 293-M7-ΔCNNM3/4 cells caused the greatest increase in $Zn^{2+}$ uptake (Fig 2A–2C). Taken together, these data indicate that both native CNNM3 and CNNM4 isoforms contribute to regulation of TRPM7 in 293-TRPM7 cells.

## PRLs stimulate TRPM7 channel function

It has been reported that CNNMs are key partners of PRLs in an evolutionarily conserved complex that regulates the intracellular $Mg^{2+}$ concentration [30,31]. PRL's association with CNNMs has been reported to increase intracellular $Mg^{2+}$ levels to stimulate cell proliferation and metastasis, although the exact mechanisms remain poorly understood [30,32]. These results motivated us to investigate whether PRLs could affect TRPM7 channel function. Overexpression of PRL-2 stimulated TRPM7-dependent $Zn^{2+}$ uptake (Fig 2F–2I), whereas overexpression of the PRL-2 harboring the R107E mutation, whose equivalent mutation in PRL-3 abolishes the binding of PRL-3 to CNNM3 [54], did not (S10 Fig). All 3 PRL isoforms are similarly capable of stimulating TRPM7 (S10 Fig). To test whether CNNMs are involved in PRL

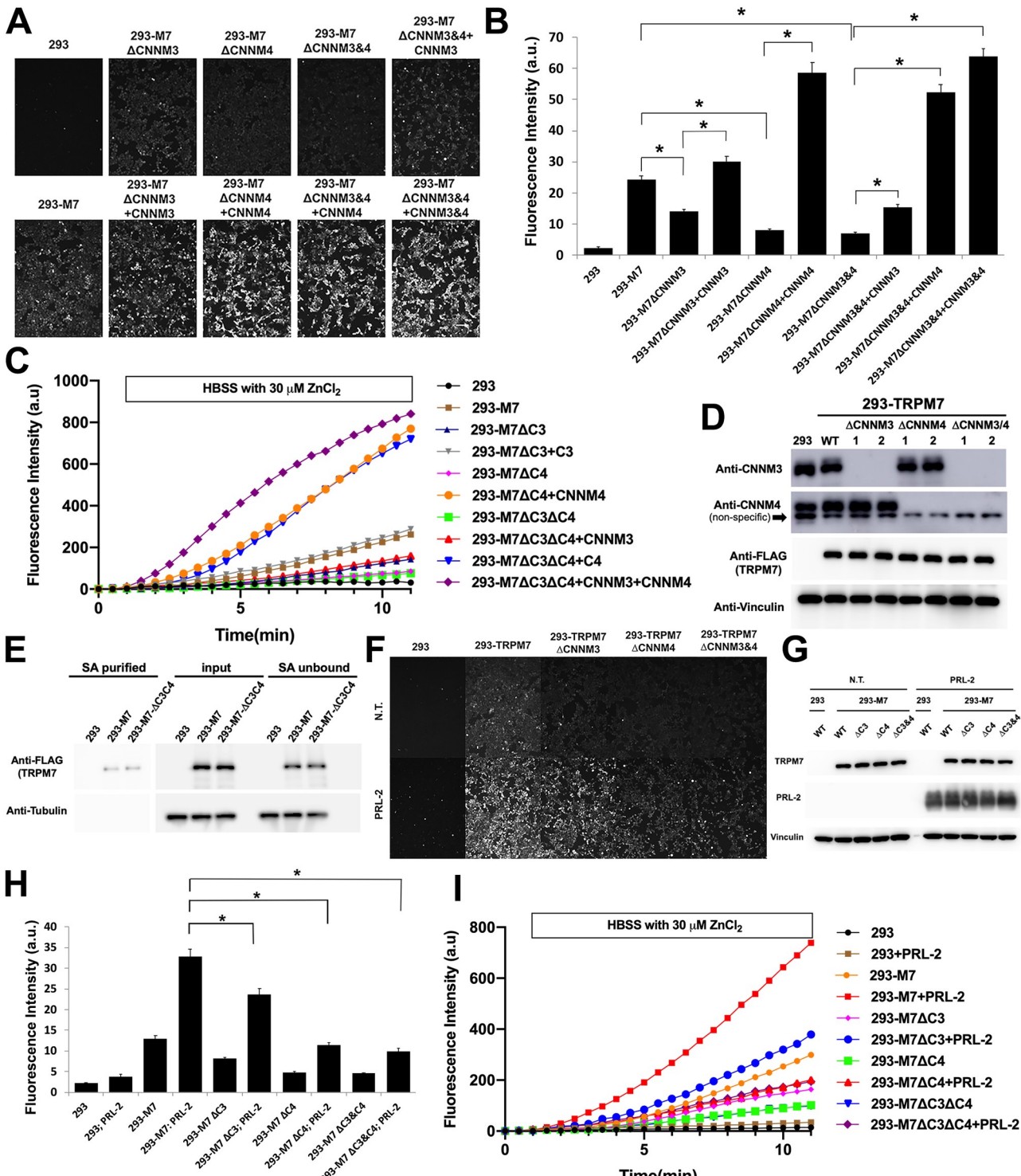

**Fig 2. CNNMs are required for TRPM7-mediated Zn²⁺ influx. (A)** A Zinc influx assay using the FluoZin-3 Zn²⁺ indicator was used to monitor TRPM7 function in intact cells under the same conditions described in Fig 1. KO of *CNNM3*, *CNNM4*, and both *CNNM3* and *CNNM4* from 293-TRPM7 cells (293-M7-ΔCNNM3, 293-M7-ΔCNNM4, and 293-M7-ΔCNNM3/4) reduced TRPM7 channel function, which could be rescued to varying degrees by reexpression of CNNM3 and/or CNNM4. The T-REx-293 cell line (293), which expresses the Tet Repressor protein, was used as the negative control. All the cells in the assay were treated with tetracycline to induce TRPM7 expression. White scale bar = 100 μM. **(B)** Quantification of the results from (A). A total of 100 cells were randomly selected for quantification. *n* = 100. * indicates a *p*-value of less than 0.05. Images taken 5 to 10 minutes after application of 30 μM ZnCl₂. **(C)** Separate time course measurements were also acquired to demonstrate that 293-TRPM7 cells lacking

CNNMs have a reduced rate of $Zn^{2+}$ influx compared to 293-TRPM7 expressing TRPM7. HBSS media was replaced with HBSS containing 30 μM $ZnCl_2$ for the period indicated. The fluorescence intensity of the cells (mean of 50 cells) were quantified for each time point. **(D)** CRISPR/Cas-9 was used to KO *CNNM3*, *CNNM4*, and both *CNNM3&4* from 293-TRPM7 cells, which express TRPM7. Western blotting of 2 independent clones showed specific KO of CNNM isoforms and similar expression of TRPM7 among the lines. Note that the antibody used to detected CNNM4 detects an unrelated protein as indicated. **(E)** Cell surface biotinylation of 293-TRPM7 WT and 293-M7-ΔCNNM3/4 cells demonstrated that surface level of TRPM7 in the 2 cell lines is similar. Shown are SA purified (SA purified) proteins, the input lysate for each cell type, and the proteins not bound to SA (SA unbound). **(F)** Zinc influx assay using the FluoZin-3 $Zn^{2+}$ indicator was used to monitor TRPM7 function in intact cells. Shown are images taken at a time point between 5 and 10 minutes after application of 30 μM $ZnCl_2$. Overexpression of PRL-2 stimulates TRPM7 channels function, which requires to varying degrees CNNM3 and CNNM4. White scale bar = 100 μM. **(G)** Western blot demonstrating expression of TRPM7 (Anti-FLAG) and PRL-2 under the conditions described in (F). **(H)** Quantification of the results from (F). A total of 100 cells were randomly selected for quantification. $n = 100$. * indicates a *p*-value of less than 0.05. **(I)** Separate time course measurements were also acquired to demonstrate that PRL overexpression stimulates an increased rate of $Zn^{2+}$ influx compared to 293-TRPM7 expressing TRPM7 and that CNNM3 and CNNM4 are required for this activity. HBSS media was replaced with HBSS containing 30 μM $ZnCl_2$ for the period indicated. The fluorescence intensity of the cells (mean of 50 cells) were quantified for each time point. Unprocessed images of blots are shown in S1 Raw Images. The underlying data for this figure can be found in S1 Data. HBSS, Hanks' balanced salt solution; KO, knockout; PRL, phosphatase of regenerating liver; PRL-2, phosphatase of regenerating liver 2; SA, streptavidin agarose.

stimulation of TRPM7 channel function, we investigated the ability to stimulate TRPM7-dependent $Zn^{2+}$ uptake in CNNM-KO cell lines. Loss of CNNM4 and CNNM3, especially CNNM4, significantly reduced the stimulation of TRPM7 channel function by PRL-2 overexpression (Fig 2F–2I). Thus, CNNMs affect the ability of overexpressed PRL proteins to stimulate the TRPM7 channel.

## CNNMs have TRPM7-dependent and TRPM7-independent activities

The functions of CNNMs are the subject of debate [42,43], with some investigators reporting CNNMs to be direct transporters that extrude $Mg^{2+}$ ions [40,41]. Other investigators have alternatively concluded that CNNMs are not directly involved in membrane transport but instead act as either intracellular $Mg^{2+}$ sensors or as $Mg^{2+}$ homeostatic mediators of other not-yet-identified transcellular transporters that alternatively regulate divalent cation influx and efflux processes [44]. To more directly evaluate the impact of KO of *CNNM3* and *CNNM4* on $Mg^{2+}$ influx, we employed a $Mg^{2+}$ uptake assay using the $^{25}Mg$ isotope. Overexpression of TRPM7 greatly stimulated $Mg^{2+}$ uptake (Fig 3A) compared to control cells. Deletion of *CNNM3* and *CNNM4* from 293-TRPM7 cells, profoundly disrupted TRPM7-mediated $Mg^{2+}$ uptake, which could be restored by reexpression of CNNM4, thus confirming TRPM7 as the conduit for CNNM-mediated $Mg^{2+}$ influx (Fig 3A). Next, we evaluated whether KO of *TRPM7* affected CNNMs ability to lower intracellular $Mg^{2+}$ levels. A similar decrease in intracellular $Mg^{2+}$ was observed in response to overexpression of CNNM2 or CNNM4 in WT HEK-293 cells compared to TRPM7-KO HEK-293 cells (293T(ΔM7)) (Fig 3B–3D), indicating that CNNM2 and CNNM4 do not require TRPM7 to lower intracellular $Mg^{2+}$cells. To directly monitor CNNM-mediated $Mg^{2+}$ influx, we took advantage of the discovery that CNNM4 requires external $Na^+$ to extrude intracellular $Mg^{2+}$ [41]. By replacing NaCl with NMDG-Cl in the external cell buffer (Hanks' balanced salt solution, HBSS), we were able to disrupt CNNM4 efflux activity and monitor CNNM4-mediated $Mg^{2+}$ influx (Fig 3E). When HBSS containing NMDG-Cl was exchanged for normal HBSS bathing HEK-293T cells overexpressing CNNM4, we were able to observe direct $Mg^{2+}$ influx as detected by an increase in Mag-Fluo-4 fluorescence. An increase in $Mg^{2+}$ influx, however, was not observed in nontransfected cells, or 293T (ΔM7) cells expressing CNNM4, or 293T WT cells overexpressing CNNM4 challenged with the TRPM7 channel inhibitor NS8593 (Fig 3E). Collectively, these data indicate that CNNMs employ TRPM7 to mediate $Mg^{2+}$influx and that the $Mg^{2+}$-lowering activities of CNNMs occur independently of the TRPM7 channel. To further rule out that CNNMs directly contribute to divalent cation influx, we took advantage of the fact that CNNM proteins have been associated

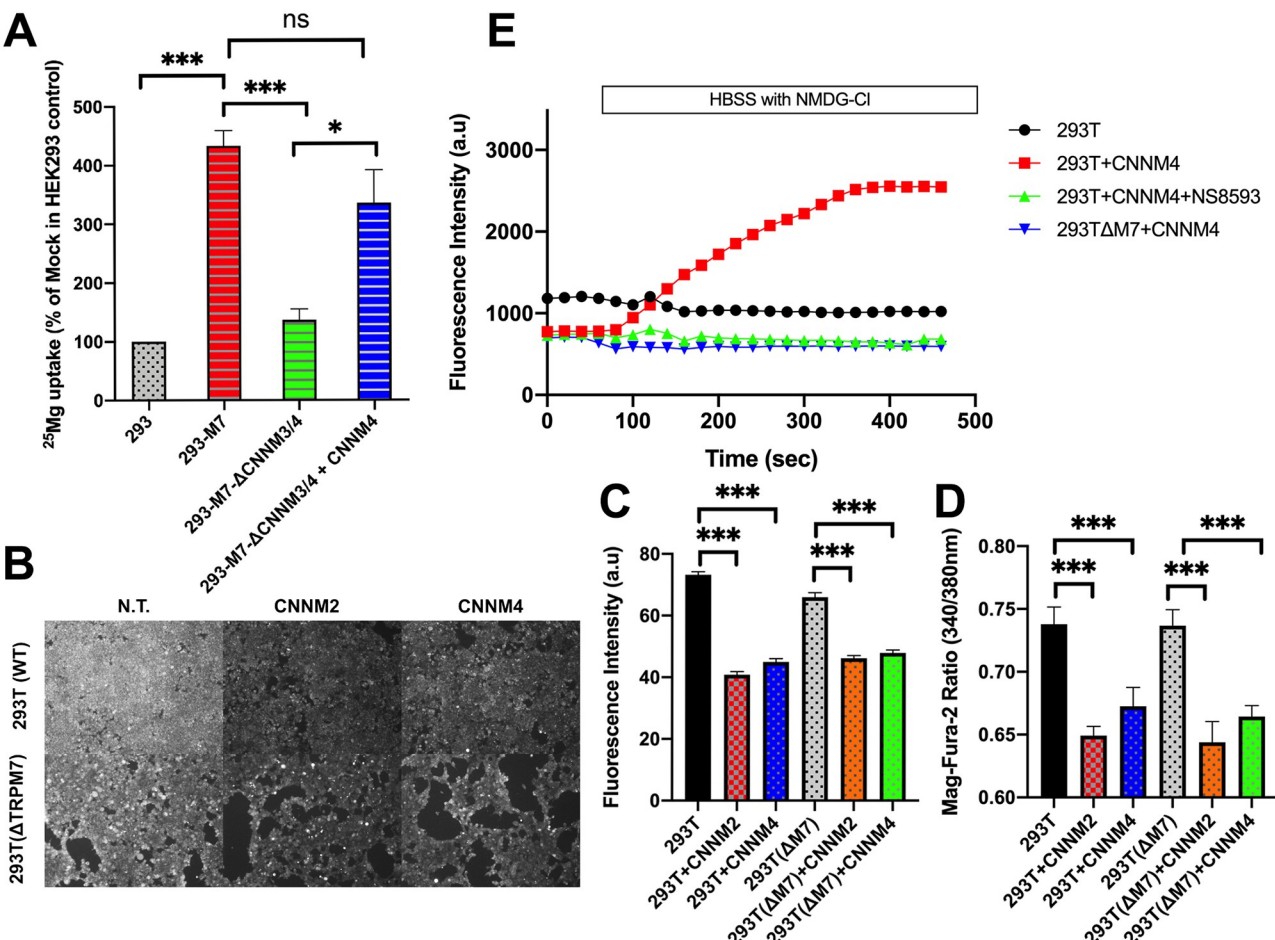

**Fig 3. CNNMs have TRPM7-dependent and independent activities. (A)** A $^{25}$Mg uptake assay was employed to assay TRPM7 channel function and its dependence on CNNMs. Moreover, 293-TRPM7 cells have significantly higher $^{25}$Mg uptake than the negative control T-REx-293 cell line (293), which expresses the Tet Repressor protein. Compared to 293-TRPM7 cells, 293-M7-ΔCNNM3/4 exhibited significantly lower $^{25}$Mg uptake, which was increased by reexpression of CNNM4. * = $p < 0.05$ and *** = $p < 0.001$, 1-way ANOVA plus Tukey post hoc. **(B)** Similar levels of free Mg$^{2+}$ were detected in HEK-293T (293T(WT)) and HEK-293T TRPM7 KO (293T(ΔM7)) cells using the Mag-Fluo4 Mg$^{2+}$ indicator. White scale bar = 100 μM. Overexpression of CNNM2 or CNNM4 substantially reduced cytosolic Mg$^{2+}$ in both cell lines to similar degrees. **(C)** Quantitation of the results from (B). $n = 50$. *** indicates a $p$-value of less than 0.001. **(D)** Mag-Fura-2 was also employed to compare the Mg$^{2+}$ levels between 293T(WT), 293T(ΔM7), and 293T(ΔM7) transfected with CNNM2 or CNNM4. Images of the cells were taken using a 510-nm filter from excitation at 340 nm versus 380 nm on an inverted Olympus IX70 fluorescence microscope, and the ratio of fluorescence intensity of the cells from excitation at the 340 nm versus 380 nm wavelengths (30 cells) was quantified. *** indicates a $p$-value of less than 0.001. **(E)** 293T(WT), 293T(WT) transfected with CNNM4, 293T(WT) transfected with CNNM4 and treated with the TRPM7 channel inhibitor NS85903 (10 μM), and 293T(ΔM7) cells transfected with CNNM4 were loaded with Mag-Fluo4. The fluorescence intensity of the cells (mean of 50 cells) were measured for 1 minute before the extracellular solution bathing the cells was replaced with a Na$^+$ free solution (replacing NaCl with NMDG-Cl) to inhibit Na$^+$-dependent Mg$^{2+}$ extrusion by CNNM4. Replacement of the buffer occurred between 50 and 100 seconds after imaging began as indicated. Overexpression of CNNM4 increased the Mg$^{2+}$ levels in 293T(WT) but not 293T(ΔM7) cells, and the increase in Mg$^{2+}$ in 293T(WT) cells caused by CNNM4 overexpression could be blocked by NS8593, which was continually present in the imaging medium and was initially added just prior to imaging. The underlying data for this figure can be found in S1 Data. WT, wild-type.

with various congenital diseases and that several disease-related missense mutants have been shown to abolish Mg$^{2+}$ efflux activity. We screened several mutant forms of CNNMs proteins that had previously been reported to disrupt Mg$^{2+}$ efflux activity indirectly (CNNM2-T568I, CNNM2ΔCBS, CNNM4ΔCBS, CNNM4-F631K, and CNNM4ΔCNBH) or directly by disrupting the putative CNNM Mg$^{2+}$binding site (CNNM2-G356A, CNNM2-E357A, CNNM4-S196P, CNNM4-S200Y, and CNNM4-N250A) [28,55,56]. Many of the tested mutants altered

both the ability of CNNM2 and CNNM4 to stimulate TRPM7-mediated $Zn^{2+}$ influx and lower intracellular $Mg^{2+}$ (S11 Fig). Interestingly, one of the CNNM $Mg^{2+}$ binding site mutants, CNNM4-S196P, which is reportedly associated with Jalili syndrome [57], completely disrupted CNNM4's capacity to lower intracellular $Mg^{2+}$ without interfering with CNNMs ability to stimulate TRPM7-dependent divalent cation influx, as monitored by our $Zn^{2+}$ influx assay (S11 Fig). Indeed, TRPM7-mediated $Zn^{2+}$ influx mediated by the CNNM4-S196P mutant was significantly higher than that produced by WT CNNM4. Thus, many of the disease-related CNNM mutations affect both CNNM-mediated divalent cation influx and cation influx activities. The fact that the CNNM4-S196P mutant disrupted the $Mg^{2+}$-lowering activity of CNNM4 without affecting CNNM-stimulated divalent cation influx, however, adds further evidence that CNNMs Mg efflux activity is independent of its divalent cation influx activity. Thus, CNNM-stimulated influx of divalent cations occurs through TRPM7, while efflux of $Mg^{2+}$ is believed to be achieved by direct binding of $Mg^{2+}$ to CNNMs, which is supported by recent structural studies of the related CorB and CorC proteins [29,56].

## CNNMs modulate TRPM7 channel activity

To more directly assess the impact of CNNMs on TRPM7 channel function, we employed electrophysiological analysis. When expressed in WT HEK-293 cells, whole-cell recordings of TRPM7 yield whole-cell currents exhibiting a significant outward current and a characteristic small inward current [4,58]. To assess the impact of CNNMs on TRPM7 conductance, we evaluated the current amplitude in 293-TRPM7 and 293-M7-ΔCNNM3/4 cells with and without reexpression of CNNM4 using an intracellular solution without $Mg^{2+}$ or Mg-ATP. Under these conditions, TRPM7's current–voltage relationship was unchanged, but TRPM7 whole-cell currents were significantly reduced in 293-M7-ΔCNNM3/4 cells compared to the parental 293-TRPM7 cells (Fig 4B and 4C). When CNNM4 was reexpressed in 293-M7-ΔCNNM3/4 cells, using stable episomal expression of CNNM4 to keep CNNM4 protein levels low (see Methods), the recorded TRPM7 currents increased. We additionally tested whether native TRPM7 would be similarly reduced by loss of *CNNM3* and *CNNM4*. To accomplish this, we knocked out CNNM3 and CNNM4 from T-REx-293 cell line, which we have previously used to express TRPM7-specific shRNAs to knockdown native TRPM7 current [47]. KO of *CNNM3 and CNNM4* together from T-REx-293 cells (293-ΔCNNM3/4) significantly reduced native TRPM7 current amplitudes compared to WT cells (293) (Fig 4F and 4G). Thus, in the absence of CNNMs, TRPM7 channel activity is severely reduced, consistent with the reduced $Zn^{2+}$ influx (Figs 2A, 2B and 4E) and $Mg^{2+}$ uptake (Fig 3A) observed in 293-M7-ΔCNNM3/4 cells. Collectively, our experiments indicate that CNNMs modulate TRPM7 channel activity and that the CNNM-TRPM7 complex constitutes a novel influx pathway for $Mg^{2+}$ as well as other divalent cations.

## Discussion

In the 1970s, Harry Rubin postulated that $Mg^{2+}$ is the key factor that regulates cell proliferation, mainly because of the importance of $Mg^{2+}$ for numerous vital cellular activities [59]. It was not until decades later that the molecular foundation for how vertebrate cellular $Mg^{2+}$ homeostasis is controlled began to be elucidated, beginning with the startling discovery that the TRPM7 ion channel is required for cell proliferation and $Mg^{2+}$ homeostasis in DT40 B cells [8]. More recently, CNNM3 and CNNM4 were found to be regulated by PRL phosphatases to increase cellular $Mg^{2+}$ levels to stimulate cancer cell growth [30–32]. By combining cell biology, biochemistry, genetic, and electrophysiology techniques, here, we have uncovered

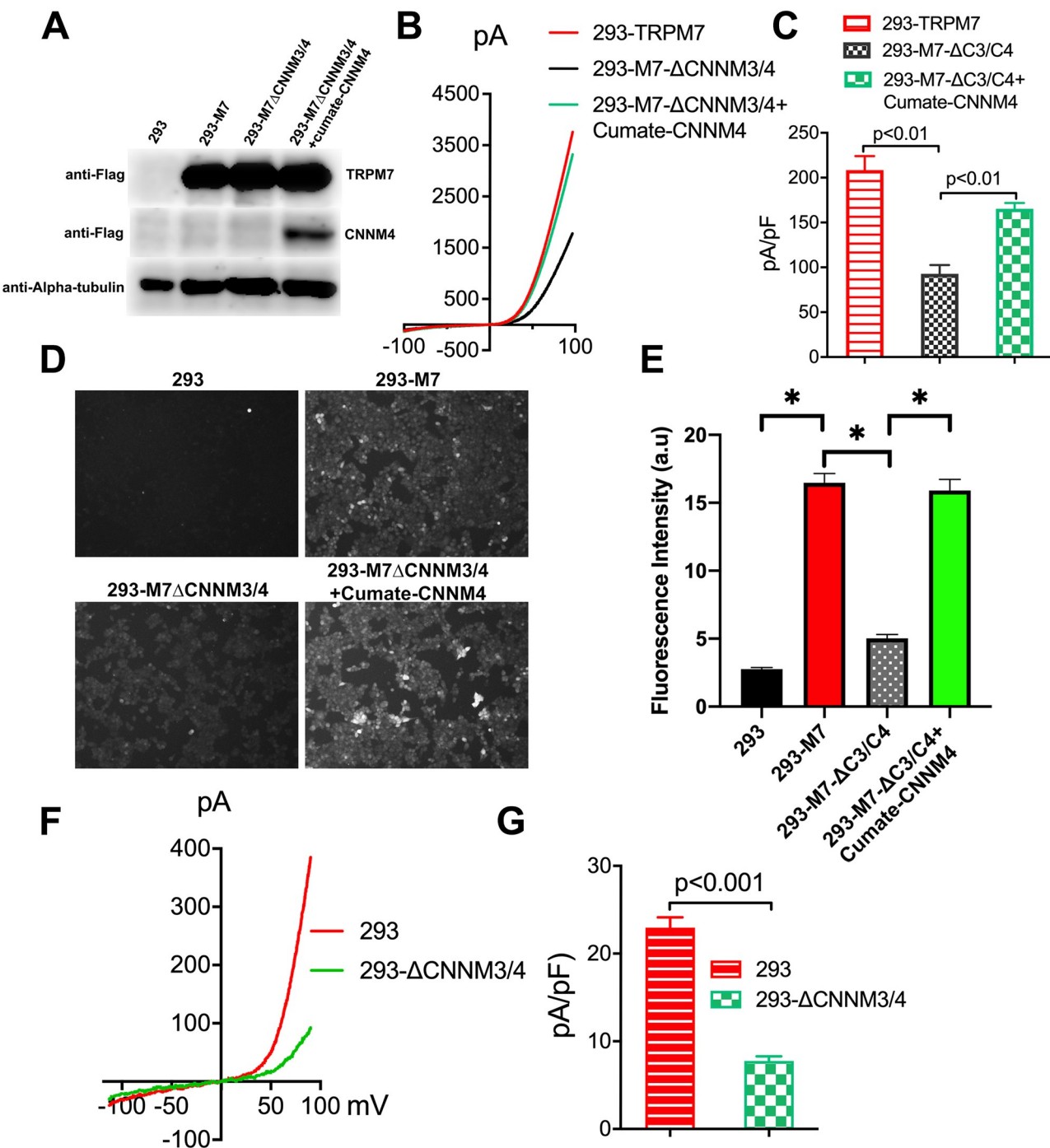

**Fig 4. Functional assessment of CNNMs on TRPM7 channel activity.** TRPM7 currents were recorded in 293-TRPM7 cells ($n = 30$), 293-M7-ΔCNNM3/4 cells ($n = 30$), and 293-M7-ΔCNNM3/4 cells ($n = 40$) with stable episomal expression of CNNM4 to keep CNNM4 protein levels low so that it would not interfere with TRPM7 protein expression. **(A)** Western blot demonstrating CNNM4 expression in the cell groups used in (B–D). **(B)** Representative TRPM7 whole-cell currents from the 3 cell lines recorded with an internal pipette solution containing 0 $Mg^{2+}$/Mg-ATP to achieve full TRPM7 channel activity. **(C)** Average current density of the different groups from (A). **(D)** Zinc influx assay using the FluoZin-3 $Zn^{2+}$ indicator was used to monitor TRPM7 function in cells from (B,C). White scale bar = 100 μM. Shown are images taken at a time point 5 to 10 minutes after application of 30 μM $ZnCl_2$. **(E)** Quantification of results from (D). A total of 100 cells were randomly selected for quantification. $n = 100$. * indicates a $p$-value of less than 0.05. **(F)** Native TRPM7 currents were recorded in the T-REx-293 Cell Line (293), which was the parental cell line of 293-ΔCNNM3/4 cells. Shown are representative TRPM7 whole-cell currents from the 2 cell lines recorded with an internal pipette solution containing 0 $Mg^{2+}$/Mg-ATP to achieve full TRPM7 channel activity. **(G)** Average current density of the different groups from (F) ($n = 16$ per group). Unprocessed images of blots are shown in S1 Raw Images. The underlying data for this figure can be found in S1 Data.

that CNNMs are bifunctional, directly regulating $Mg^{2+}$ efflux, while at the same time capable of employing the TRPM7 channel to stimulate divalent cation entry into cells.

Our experiments indicate that CNNMs have both $Mg^{2+}$ import and export activities, with CNNM-mediated $Mg^{2+}$ import occurring when CNNM proteins assemble into macromolecular complexes with the TRPM7 ion channel. In the absence of TRPM7, CNNMs remain capable of lowering intracellular $Mg^{2+}$, which a recent structural study reports occurs through direct physical binding of the cation to CNNMs transmembrane domain [56]. Thus, CNNMs regulate opposing activities, acting to extrude $Mg^{2+}$ out of the cells and, as our data have shown, functioning in complex with TRPM7 to stimulate cellular uptake of $Mg^{2+}$ as well as other divalent cations. These data point to a model in which CNNMs function as a signaling hub through which cellular $Mg^{2+}$ homeostasis is maintained through simultaneous regulation of both cellular $Mg^{2+}$ influx and $Mg^{2+}$ efflux activities, with $Mg^{2+}$ entry occurring through TRPM7. Mg-ATP, PRLs, and ARL15, which bind to CNNMs, likely regulate both influx and efflux activities of CNNMs. For example, PRLs were previously reported to prevent CNNM4-dependent $Mg^{2+}$ efflux activity [30]. Our experiments have revealed that PRL proteins stimulate TRPM7-mediated divalent cation influx into cells. Collectively, these results suggest that increases in cellular $Mg^{2+}$ may be achieved through inhibition of CNNM-dependent $Mg^{2+}$ efflux and stimulation of divalent cation influx through TRPM7. Interestingly, a recent study demonstrated that expression of PRL proteins is affected by external $Mg^{2+}$ availability [60], with increases in PRL mRNA levels occurring when external $Mg^{2+}$ levels are decreased. Expression of TRPM7 mRNA has also been reported to be regulated by $Mg^{2+}$ [61], potentially through regulated translation of the full-length protein [62]. Thus, changes in $Mg^{2+}$ availability could influence cellular magnesium homeostasis by regulating protein levels of PRLs and TRPM7 [60,61]. Adding further complexity to CNNM and TRPM7 regulation is the protein ARL15, which was recently shown to bind to CNNMs and influence its glycosylation [33]. Overexpression of ARL15 in renal carcinoma cells decreased cellular $Mg^{2+}$ uptake, whereas KO of the protein increased it [33]. While our manuscript was under review, a preprint appeared on the bioRxiv server, reporting that macromolecular complexes of TRPM7, CNNMs, PRLs, and ARL15 can be detected in rodent brain [63]. The article, which was recently accepted for publication, demonstrated that heterologous overexpression of ARL15 with TRPM7 in HEK-293 cells suppressed TRPM7 channel activity [64]. Using the $Zn^{2+}$ influx assay, we similarly found that overexpression of ARL15 inhibited TRPM7-mediated $Zn^{2+}$ influx (S12 Fig). Conversely, we also find that knockdown of ARL15 in HEK-293 cells increased TRPM7-mediated $Zn^{2+}$ influx (S12 Fig). Thus, PRLs and ARL15 appear to have opposing actions on TRPM7 function. Although these proteins' role in the regulation of CNNM-TRPM7 still remain poorly understood, the present data suggests that in vivo regulation of CNNM-TRPM7 is complex and is likely subject to many layers of regulation, which could vary depending on the expression patterns and levels of the individual proteins in different cells and organ systems.

Indeed, our experiments have demonstrated that native CNNMs can regulate endogenous TRPM7 channel function in nonpolarized cells (Fig 4F and 4G). We also discovered that a complex between the proteins can be detected in multiple cell types, including the endogenous TRPM7 and CNNM4 proteins in ZR-75-1 breast cancer cells (S3 Fig). An intriguing question for future investigation is whether TRPM7 and CNNM proteins function together in polarized epithelial cells. In this cell type, TRPM7 is presumed to localize to the apical membrane, whereas CNNMs have been reported to localize to basolateral membranes [30,40,65–67]. Whether CNNMs have the potential to also localize to apical membranes of polarized epithelial cells has not been extensively investigated. We therefore examined the capacity of CNNM1-4s to localize to apical versus basolateral membranes in OK cells, a proximal tubule epithelial cell

line. To varying degrees, CNNMs could be detected on both apical and basolateral membranes of OK cells, with CNNM2 more easily found on the apical membrane than CNNM4 (S13 Fig). CNNM1 and CNNM3 could be detected in both membrane compartments of OK cells. A similar trend for the cellular distribution of CNNM2 and CNNM4 was also observed when the 2 proteins were expressed in other polarized epithelial cell lines, including Caco-2 (S14 Fig), and immortalized mouse primary proximal tubule epithelial cells (RPTEC/TERT1) (S15 Fig). These experiments indicate that CNNMs, to varying degrees, have the potential to localize to apical membranes where TRPM7 is believed to function. While additional studies are required to investigate the potential function of a CNNM-TRPM7 complex in polarized epithelial cells, existing data already suggest in some cell types the 2 proteins can found together in specialized compartments. For example, CNNM4 is found expressed in TRPM7-positive odontoblasts, where both TRPM7 and CNNM4 are localized to the odontoblastic process [68].

Last, our experiments support a model in which CNNMs can function as a regulatory subunit of the TRPM7 channel. As far as we know, despite the importance of the TRP channel family with approximately 28 members in vertebrates, very few of these channels have been shown to possess regulatory subunits, which is rare among ion channel families. Immunoprecipitation experiments indicate a robust interaction between TRPM7 and CNNM proteins. Interestingly, CNNMs also bind to TRPM6, suggesting that CNNMs could potentially influence $Mg^{2+}$ influx through TRPM6 homomeric or TRPM6/TRPM7 heteromeric channels, although this remains to be investigated. Conversely, our results suggest that the composition of CNNM-TRPM7 complexes may also be diverse in vivo. In HEK-293 cells, KO of *CNNM3* and *CNNM4* significantly reduced TRPM7-dependent divalent cation influx and whole-cell currents, without affecting TRPM7 protein or channel surface levels. These data indicate that both native CNNM3 and CNNM4 are involved in regulating TRPM7 channel in HEK-293 cells, raising the question of whether they do so individually or together as a complex. Interestingly, we find that CNNMs are capable of forming heteroligomers, with native CNNM3 and CNMM4 forming a complex together (S16 Fig). The function of CNNM heteroligomers has not been investigated and whether they cooperatively regulate TRPM7 remains to be determined in future work. However, the discovery that TRPM7 has the potential to interact with multiple CNNM isoforms in vitro, as well as in native rodent brain [64], has uncovered a potential mechanism by which diverse control of the channel could be achieved in different tissues and cell types by varying the isoforms that makeup the CNNM-TRPM7 complex. We speculate that given $Mg^{2+}$ homeostasis profound impact on so many essential cellular activities, from cell metabolism to cell proliferation, and TRPM7's channel influence on disease events from cancer to stroke, CNNM-TRPM7 complexes of varying composition will likely be found contributing to diverse physiological and pathological processes. We conclude that CNNMs are dual function proteins, mediating $Mg^{2+}$ efflux, while at the same time functioning as novel homeostatic factors of TRPM7 to stimulate the influx of $Mg^{2+}$ and other divalent cations through the channel. Future work will be devoted to unraveling how the activities of CNNM-TRPM7 complexes and their associated binding proteins are controlled and employed in vivo.

## Methods

### Plasmids and adenovirus constructs

Expression plasmids for FLAG-tagged human CNNM1, mouse CNNM2, human CNNM3, and human CNNM4 in the pCMV-4A were previously described [28] and were a kind gift of Dr. Hiroaki Miki (Osaka University, Japan). Expression plasmids for Myc-DDK tagged human PRL-1, PRL-2, and PRL-3 in the pCMV6-Entry vector was purchased from Origene (Rockville, Maryland, United States of America). The PRL-2(R107E) mutant was made using

Quikchange (Agilent, Santa Clara, CA, USA), following the manufacturer's instructions; primers are available upon request. The 3xHA-TurboID-NLS_pcDNA3 was a gift from Alice Ting (Addgene plasmid # 107171; http://n2t.net/addgene:107171; RRID:Addgene_107171). To create pCuO-MCS-CNNM3-FLAG-TurboID, COOH-terminal FLAG-tagged human CNNM3 and the TurboID coding sequence from 3xHA-TurboID-NLS_pcDNA3 was subcloned into the CuO-MCS vector (System Biosciences, Mountain View, California, USA), which allows for cumate-inducible sustained transgene expression in cells, using the Cold Fusion Cloning Kit (System Biosciences). The primers to introduce CNNM3 by PCR from the pCMV4A vector were 5′-CAT CGC GAC GTT TAA ACA TGG CGG CGG CGG TAG CT-3′ and 5′-CAG CCG GTG GTC TTC TCG TCC ATC TTA TCG TCG TCA TCC TTG TAA TCC TCG AG-3′. The primer sequences to introduce TurbioID from 3xHA-TurboID-NLS_pcDNA3 were 5′-CTC GAG GAT TAC AAG GAT GAC GAC GAT AAG AAA GAC AAT ACT GTG CCT-3′ and 5′-GGC GAT CGC TGT ACA GCA TGC CTA CTT TTC GGC AGA CCG CAG ACT GAT-3′. Similarly, pCuO-MCS-CNNM4-FLAG was created by PCR and the Cold Fusion Cloning Kit. The primers used were 5′-GGC CGG CCA TCG CGA CGT TTA AAC ATGG CGC CGG TGG GCG GGG GC-3′ and 5′-GGC GAT CGC TGT ACA GCA TGC CTA CTT TTC GGC AGA CCG CAG ACT GAT-3. To create pcDNA6-EFYP-FLAG-TurboID, EYFP (Takara Bio USA, San Jose, California, USA) was first cloned into pcDNA6-V5-HisB vector in frame in the NheI and HindIII sites using the following primers: 5′-CAT CAT GCT AGC CTA CCG GTC GCC ACC ATG GTG-3′ and 5′-CAT CAT AAG CTT GTC CAT GCC GAG AGT GAT CCC-3′. Next, the TurboID sequence from 3xHA-TurboID-NLS_pcDNA3 was subcloned by PCR, to introduce an $NH_2$-terminal FLAG tag and the TurboID, into the pcDNA6-V5-HisB vector containing EYFP using the following primers: 5′- CAT CAT CAT GGA TCC ATG GAC TAC AAG GAC GAC GAT GAC AAG AAA GAC AAT ACT GTG CCT-3′ and 5′-CAT CAT CAT GAA TTC CTA CTT TTC GGC AGA CCG CAG ACT GAT-3′. Recombinant adenoviruses expressing FLAG-CNNM2 (Ad-FLAG-CNNM2) and FLAG-CNNM4 (Ad-FLAG-CNNM4) were made using the ViraPower Adenoviral Expression system (Invitrogen, California, USA), following the manufacturer's instructions. First, FLAG-CNNM2 and FLAG-CNNM4 were subcloned into the pENTR-D vector using the following forward primers, respectively, 5′-CAC CAT GGC GCC GGT GGG CGG GGG C-3′ and 5′-CAC C ATGATTGGCTGTGGCGCTTGTGA ACCCG-3′ and the same reverse primer 5′-GGC GAT CGC TGT ACA GCA TGC CTA CTT TTC GGC AGA CCG CAG ACT GAT-3. FLAG-CNNM2 and FLAG-CNNM4 in the pENTR vectors were then was then transferred to the destination vector pAd/CMV/V5-DEST using the Gateway system. pAd/CMV/V5-SR-TRPM7 recombinant adenovirus (Ad-TRPM7-HA), which expresses HA-tagged mouse TRPM7, was previously described [69]. pcDNA5-FRT-TO-HA-TRPM7 was previously described [47]. pcDNA5-FRT-TO-HA-TRPM6 was a gift of Dr. Lixia Yue (UCONN Health Center). We engineered mutation in mouse CNNM2 and human CNNM4 in the pCMV-4A vector (CNNM2-T568I, CNNM2ΔCBS(469–578), CNNM4ΔCBS(396–505), CNNM4-F631K, CNNM4ΔCNBH(512–775), CNNM2-G356A, CNNM2-E357A, CNNM4-S196P, CNNM4-S200Y, and CNNM4-N250A) using site-directed mutagenesis.

## Cell lines

All cells were maintained in a Dulbecco's Modified Eagle Medium (DMEM), high glucose media with 10% FBS in a humidified 37°C, 5% $CO_2$ incubator unless otherwise specified below. The 293T cell line was purchased from ATCC (CRL-3216, Manassas, Virginia, USA). Moreover, 293T cells deficient in TRPM7 (29T3-ΔM7) were previously described [70] and were grown in DMEM with 10% FBS supplemented with 10 mM $MgCl_2$ and were a gift of

Dr. David Clapham (HHMI, Janelia Research Campus). The LTRPC7 cell line [4], herein referred to as 293-TRPM7 cells, expresses FLAG-tagged mouse TRPM7 under tetracycline control and was generously provided by Dr. Andrew Scharenberg (University of Washington). To avoid cell detachment due to TRPM7 overexpression in the functional studies, LTRPC7 (293-TRPM7) cells were plated on polylysine coated plates or coverslips. The LTRPC7 cells were used for all the functional studies, with the exception of the $Zn^{2+}$ imaging experiments conducted in S6 Fig, which were conducted using mouse HA-tagged TRPM7 WT (293-M7(WT)) and mouse HA-tagged TRPM7-E1047K (293-M7(E1047K)) cell lines that we developed in previous studies [47,51]. Furthermore, 293-M7(WT) and 293-M7 (E1047K) express HA-tagged mouse TRPM7 and the E1047K mutation specifically targets the pore of the channel, rendering it inactive [51]. WT HAP1 cells and TRPM7-deficient HAP1 cells were a kind gift of Dr. Vladimir Chubanov and have been previously described [6]. HAP1 cells were maintained in improved minimal essential media (IMEM) with 10% FBS and TRPM7-deficient cells were maintained in IMEM with 10% FBS containing 10 mM $MgCl_2$. Hela cells were a gift of Dr. Stephen Zheng (Rutgers University). ZR-75-1 cells were a gift from Dr. Leroy Liu (Rutgers University). LLC-PK1 cells were a gift of Dr. Goufeng You (Rutgers University). The OK cell line was provided by Dr. Judith A. Cole (Department of Biological Sciences, University of Memphis). The OK cells were cultured in DMED/F12 medium supplemented with 5% of FBS. RPTEC/TERT1 cells were from ATCC (CRL-4031) and were grown according to the manufacturer's instructions.

## CNNM3 and CNNM4 knockout cell lines

The LTRPC7 cells lines expressing TRPM7 and deficient in *CNNM3*, *CNNM4*, or *CNNM3* and *CNNM4* (293-M7-ΔCNNM3, 293-M7-ΔCNNM4, and 293-M7-ΔCNNM3/4 cells) were created using the Alt-R CRISPR/Cas-9 System from Integrated DNA Technologies (Coralville, Iowa, USA). The T-REx-293 Cell Line (Thermo Fisher Scientific, Waltham, Massachusetts, USA), which is a Flp-In cell line that expressed the Tet Repressor protein, was used to create 293-ΔCNNM3/4 cells. The Alt-R CRISPR/Cas-9 guide RNAs for CNNM3 and CNNM4 were 5′-AltR1/rArUrGrGrUrUrGrUrArGrArArArArCrGrArGrUrGrArGrUrUrUrUrArGrArGrCrUrArUrGrCrU/AltR2/-3′ and 5′-/AltR1/rCrArArGrUrCrGrUrGrUrGrGrGrArCrGrArArArCrCrGrUrUrUrUrArGrArGrCrUrArUrGrCrU/AltR2/-3. The guide RNAs were transfected into LTRPC7 cells and T-REx-293 cells with Alt-R CRISPR crRNA, Alt-R CRISPR/Cas-9 tracrRNA and recombinant Alt-R S. pyogenes Cas9 Nucleus V3. Two days following transfection, the cells were serially diluted. Colonies were screened for loss of protein expression using antibodies specific for CNNM3 (NBP2-32134, Novus Biologicals Centennial, Colorado, USA) and CNNM4 (ab191207; Abcam Cambridge, UK). All of the CNNM KO cell lines produced were cultured in DMEM with 10% FBS and did not require $Mg^{2+}$ supplementation to sustain cell proliferation. The genotypes of individual cell lines were further confirmed by PCR amplification of genomic DNA to verify targeting of the targeted locuses. Moreover, 293-M7-ΔCNNM3 cells have a deletion of "ACTC" from nucleotides 1,147 to 1,150 of the consensus CDS for CNNM3 (CCDS2025. 1). Furthermore, 293-M7-ΔCNNM3/4 cells have a deletion of "ACTCGTTTCTACAACCATC" from nucleotides 1,147 to 1,165 of the consensus CDS for CNNM3 (CCDS2025.1). For both 293-M7-ΔCNNM4 and 293-M7-ΔCNNM3/4, there is a deletion of "CGAA" from 179–182" from nucleotides 1,147 to 1,150 of the consensus CDS for CNNM4 (CCDS 2024.2). In addition, 293-ΔCNNM3/4 cells have a deletion of "ACTCGTTTCTACAACCATC" from nucleotides 1,147 to 1,165 of the consensus CDS for CNNM3 (CCDS2025.1); these cells also have a deletion of "CGAA" from 179 to 182" from nucleotides 1,147 to 1,150 of the consensus CDS for CNNM4 (CCDS 2024.2).

## Immunoprecipitation and detection of protein

For analysis of protein expression, cells were lysed with RIPA buffer (50 mM TRIS (pH 7.4), 150 mM NaCl, 1% Igepal 630, 1% sodium deocycholate, and 0.1% SDS) containing protease inhibitors. For immunoprecipitation experiments, cells were lysed in lysis buffer (50 mM TRIS (pH 7.4), 150 mM NaCl, and 1% Igepal 630). All biochemical procedures were conducted on ice or at 4°C. Membrane proteins were solubilized by incubating the lysis mixture at 4°C for 30 minutes to ensure effective solubilization of TRPM7. This was verified in control experiments by centrifuging lysates for 45 minutes at $100,000 \times g$ and analyzing for the presence of TRPM7 in the supernatant by SDS-PAGE and western blotting. The proteins were resolved by SDS-PAGE and western blotting using standard protocols. For the immunoprecipitation experiments, 10-cm dishes of cells were lysed with 1 ml of lysis buffer, and the samples were then cleared by centrifugation at $15,600 \times g$ for 10 minutes. Supernatants (lysates) were subjected to immunoprecipitation as follows. HA-TRPM7 proteins were immunoprecipitated overnight using 40 μL of HA-agarose (Roche Life Sciences, Indiana, USA). FLAG-TRPM7 proteins were immunoprecipitated overnight using 40 μL of M2 FLAG-agarose (Sigma-Aldrich, Missouri, USA). Native TRPM7 was immunoprecipitated overnight using 2 μg Anti-TRPM7 antibody from Alomone Labs (#ACC-047; Jerusalem, Israel) bound to 40 μL Protein A agarose (Santa Cruz Biotechnology, Dallas, Texas, USA). For the immunoprecipitation experiments, beads were washed 3 times with lysis buffer, the bound proteins were eluted with 40 μL Laemmli sample buffer, and the protein resolved by SDS-PAGE and western blotting using standard procedures.

Anti-β-actin (sc-47778; Santa Cruz Biotechnology) was used to detect β-actin as loading control. Similarly, antibodies targeting vinculin and alpha-tubulin (Sigma-Aldrich) were used to monitor protein loading for western blotting. The Anti-FLAG M2 antibody (Sigma-Aldrich) was used to detect FLAG-TRPM7. The rat monoclonal antibody Anti-HA antibody (clone 3F10, Roche Life Sciences) was used to detect HA-tagged TRPM7. The TRPM7-specific PLIKC47 antibody (α-C47), which recognizes residues 1,816 to 1,863 from rat TRPM7, has been previously described [46]. Overexpressed PRL-2 was detected using the Anti-PRL-2 antibody, clone 42 (Millipore Sigma, Darmstadt, Germany), or Anti-FLAG M2 antibody (Sigma-Aldrich). ARL15 was detected using the Anti-ARL15 antibody (Proteintech, Rosemont, Illinois, USA). Knockdown of ARL15 was achieved using Silencer Select Predesigned siRNAs targeting ARL15 (siRNA IDs: s29264 and s29266) compared to Silencer Select Negative Control #1 siRNA (Thermo Fisher Scientific). Transient transfection of siRNAs was accomplished using LipofectamineRNAiMAX at a 10 nM final concentration for 48 hours following the manufacturer's instructions. Transient transfection of DNA was conducted using TurboFect Transfection Reagent (Life Technologies, Grand Island, New York, USA), or Lipofectamine 3000 (Themo Fisher Scientific). For the coimmunoprecipitation experiments in Fig 1, HEK-293T cells were transiently transfected. pcDNA5-FRT-TO-HA-TRPM7 and pCMV-4A-FLAG-CNNM1-4 constructs were used. For the coimmunoprecipitation experiments in S1 Fig, pcDNA5-FRT-TO-HA-TRPM7 and pCMV-4A-FLAG-CNNM1-4 constructs were used. For the coimmunoprecipitation experiments in S3 Fig, Ad-HA-TRPM7 was employed to virally express HA-TRPM7 in OK, HAP1, and RPTEC/TERT1 cells. For the coimmunoprecipitation experiments involving Hela cells, pcDNA5-FRT-TO-HA-TRPM7 was transiently transfected into these cells.

## Immunocytochemistry

Cells were seeded onto polylysine-coated glass coverslips placed into 24-well plates and then transfected or virally transduced as specified in the figure legends. Cells were fixed using 4% paraformaldehyde in PBS for 30 minutes at room temperature. Moreover, 0.1% Triton X-100

in PBS was used to permeabilized the cells at 25˚C for 5 minutes, and then 5% FBS in PBS were used for blocking at 30˚C for 1 hour. Next, cells were incubated in appropriate primary antibodies for 1 hour at 30˚C. To detect HA-TRPM7, we used either a rat monoclonal antibody anti-HA antibody (3F10; Roche Life Sciences) or a rabbit monoclonal antibody anti-HA (C29F4) from Cell Signaling Technology (Danvers, Massachusetts, USA), depending on the experiment. To detect NHERF-1, we used the NHERF-1 mouse monoclonal antibody (sc-271552) from Santa Cruz Biotechnology. To detect the $Na^+/K^+$-ATPase, we used the mouse monoclonal antibody against $Na^+/K^+$-ATPase (sc-21712) from Santa Cruz Biotechnology. For detection of overexpressed FLAG-CNNM proteins, we employed either the M2 anti-FLAG antibody mouse monoclonal antibody (F1804) from Sigma-Aldrich or the anti-FLAG rabbit monoclonal antibody (D6W5B) from Cell Signaling. After three 5-minute washing steps with 5% FBS in PBS, Alexa Fluor 488 and Alexa Fluor 568 were used as secondary antibodies (Thermo Fisher Scientific), with DAPI (Sigma-Aldrich) used to stain the nucleus. After 3 washes with 5% FBS in PBS and 3 washes with PBS, the coverslips with cells were mounted onto glass slide with Aqua Poly/Mount (Polyscience, Warrington, Pennsylvania, USA). Images were taken by a Yokogawa CSUX1-5000 microscope under 63× magnification using 488 nm and 561 nm wavelengths at the Rutgers RWJMS CORE Confocal facility.

## Mass spectrometry

We conducted mass spectrometry analysis of TRPM7 purified from the LTRPC7 cell line, described above, grown intentionally in the absence of tetracycline to keep FLAG-TRPM7 protein levels as low as possible and at equilibrium inside the cells. A 10-cm dish of LTRPC7 cells without tetracycline was used for purification of FLAG-TRPM7. A 10-cm dish of HEK-293T cells were used as a negative control. The experiment under the same conditions was repeated once to give 2 biological replicates. Cells were lysed with 1 ml of lysis buffer (50 mM TRIS (pH 7.4), 150 mM NaCl, and 1% Igepal 630) and were solubilized by incubating the lysis mixture at 4˚C for 1 hour. LTRPC7 and control HEK-293-T cell lysates were then subjected to immunoprecipitation overnight using 40 μL of M2 FLAG-agarose. The beads were washed 3 times with PBS with 1% Tween-20, the bound proteins were eluted with 40 μL Laemmli sample buffer. Immunoprecipitation of FLAG-TRPM7 was verified by SDS-PAGE and western blotting using standard procedures. For mass spectrometry analysis, we employed a protocol we previously employed [45]. The samples were resolved in a Bis-Tris polyacrylamide gel. The excised gel band was subjected to in-gel reduction, alkylation, tryptic digestion, and peptide extraction with a standard protocol. Peptides were solubilized in 0.1% trifluoroacetic acid and analyzed by Nano LC-MS/MS with Dionex Ultimate 3000 RLSC Nano System interfaced with a Velos-LTQ-Orbitrap (Thermo Fisher Scientific) or a QExactive HF (Thermo Fisher Scientific) based on instrument availability. Samples were loaded onto a self-packed 100 μm × 2 cm trap (Magic C18AQ, 5μm 200 Å; Michrom Bioresources, Pennsylvania, USA) and washed with Buffer A (0.2% formic acid) for 5 minutes with a flow rate of 10 μl/min. We employed a trap brought in line with the analytical column (Magic C18AQ, 3 μm 200 Å, 75 μm × 50 cm; Michrom Bioresources). Peptides were fractionated at 300 nL/min using a segmented linear gradient 4% to 15% Buffer B (0.2% formic acid in acetonitrile) in 35 minutes, 15% to 25% Buffer B in 65 minutes, and 25% to 50% Buffer B in 55 minutes. Mass spectrometry data were acquired using a data-dependent acquisition procedure with a cyclic series of a full scan acquired with a resolution of 60,000 (Velos-LTQ-Orbitrap) or 120,000 (QExactive HF) followed by MS/MS of the 20 most intense ions and a dynamic exclusion duration of 30 seconds for both instruments. Proteome Discoverer was used for database search and analysis. Data were searched against most updated SwissProt database using MASCOT (v 2.3). Precursor ion mass error tolerance was set

to ±10 ppm (Velos-LTQ-Orbitrap) or +/− 7 ppm (QExactive HF) and fragment mass error tolerance to ±0.4 Da (Velos-LTQ-Orbitrap) and +/− 20 ppm (QExactive HF). Cysteine carbamidomethylation was set as a complete modification, acetylation on N-terminus of protein, oxidation on methionine, phosphorylation on serine, and threonine and tyrosine were set as variable modifications. Site localization was analyzed using PTMRS. Only spectra of high confidence (false discovery rate [FDR] were set at 0.01 for PSM) were reported [45]. Mass spectrometry proteomics data have been deposited to the ProteomeXchange Consortium via the PRIDE partner repository with the dataset identifier PXD026635. S1 Table shows the proteins that were identified from the LTRPC7 cells but not the control cells in both experimental samples.

## Cell surface biotinylation

A 10-cm dish of cells were biotinylated for 30 minutes at 25°C in 0.5 mg/mL sulfo-NHS-LC-LC-biotin (Thermo Fisher Scientific). Cells were washed with PBS 3 times and then lysed in RIPA buffer. Moreover, 10% (v/v) of the sample was taken as input control, and the rest of the protein lysates were incubated overnight with Pierce Streptavidin Agarose (Thermo Fisher Scientific). The next day, unbound protein was collected, and the beads were washed 3 times with lysis buffer. The collected samples were subsequently subjected to SDS-PAGE and western blotting.

## Proximity labeling

We employed the method developed by Branon and colleagues for proximity labeling using CNNM3 [49]. Briefly, pCuO-MCS-CNNM3-FLAG-TurboID or pcDNA6-EFYP-FLAG-TurboID, functioning as a negative control, were transiently transfected into the LTRPC7 cell line [4], which expresses FLAG-tagged mouse TRPM7 under tetracycline control. Biotin at a concentration of 500 μM was added to extracellular media for 10 minutes in initiate proximity labeling, followed by washing of the cells with PBS solution and cell lysis using RIPA buffer. Biotinylated proteins were purified using Pierce Streptavidin Agarose (Thermo Fisher Scientific) and labeling of FLAG-TRPM7 was assessed by SDS-PAGE and western blotting using the ANTI-FLAG M2 antibody.

## Zinc influx assay

We adapted a protocol from the one employed by Inoue and colleagues to measure $Zn^{2+}$ influx by TRPM7 [71]. Cells were plated into 24-well dishes coated with polylysine to aid comparison between individual samples by fluorescence microscopy. Before labeling, cells were washed with HBSS containing 0.137 M NaCl, 5.4 mM KCl, 0.25 mM $Na_2HPO_4$, 6 mM glucose, 0.44 mM $KH_2PO_4$, 1.3 mM $CaCl_2$, 1.0 mM $MgSO_4$, and 4.2 mM $NaHCO_3$. Cells were then labeled with the Zinc indicator FluoZin-3 (2.5 μM) in HBSS following manufacturer instructions (Thermo Fisher Scientific). The cells were then washed once with HBSS and then placed back into HBSS before images were quickly acquired on an inverted Olympus IX70 fluorescence microscope with a 10X phase contrast objective (Olympus Neoplan 10/0.25 Ph, Olympus, Tokyo, Japan). Cells were visually inspected for uneven dye loading prior to imaging. To stimulate $Zn^{2+}$ influx, $ZnCl_2$ (30 μM $ZnCl_2$) in HBSS was introduced to cells. For static measurements, images of cells from the different samples were taken at a specific time point 5 to 10 minutes after the addition of 30 μM $ZnCl_2$. In control experiments, uneven incorporation of the probe was not observed, and cells not challenged 30 μM $ZnCl_2$ had an extreme low fluorescence compared to unlabeled cells. For time course measurements, images of cells were acquired on the inverted Olympus IX70 fluorescence microscope every 30 seconds. Moreover,

30 μM $ZnCl_2$ was added between 40 and 50 seconds after imaging began in most experiments. Fluorescence intensity from individual cells were quantified using Fiji [72].

## Measurement of intracellular $Mg^{2+}$

We used Mag-Fluor-4 and Mag-Fura-2 (Thermo Fisher Scientific) to assess differences in intracellular $Mg^{2+}$ levels between different cell samples. Cells were plated in 24-well dishes to aid comparison between individual samples. For experiments with Mag-Fluo-4, cells in the 24-well dish were labeled with the $Mg^{2+}$ indicator at a concentration of 2.5 μM following manufacturer instruction (Thermo Fisher Scientific). The cells were then washed twice with HBSS and then incubated with HBSS and images from the different samples were quickly acquired on an inverted Olympus IX70 fluorescence microscope with a 10X phase contrast objective (Olympus Neoplan 10/0.25 Ph). Fluorescence intensity was quantified (50 or 100 cells per field) using Fiji [72]. For experiments using Mag-Fura-2, the Olympus IX70 microscope was equipped with a FURA filter set, including Dual exciter ER340ex—ET380ex—400lpx4–510/80m, a filter cube, a filter wheel with internal shutter, and an upgraded filter/shutter control card installed into a refurbished LUDL controller system. Cells were plated into 24-well dishes with a glass bottom plate coated with polylysine. Cells were labeled with Mag-Fura-2 (4 μM) and BAPTA-AM (25 μM) to reduce any contribution of intracellular free $Ca^{2+}$ to the measurements, in a modified Tyrode solution containing 10 mM Hepes, 110 NaCl, 5 mM $NaHCO_3$, 5 mM KCl, 1.8mM $CaCl_2$, 1.0 mM $MgCl_2$, 1.0 mM $NaH_2PO_4$, 5 mM Glucose, 10 mM sodium pyruvate, and MEM nonessential amino acids solution (10 mM glycine, alanine, asparagine, aspartic acid, glutamic acid, proline, and serine). Cells were washed once with the Tyrode solution and then imaged in the same buffer. Images of the cells were taken using a 510 nm filter from excitation at 340 nm versus 380 nm on the inverted Olympus IX70 fluorescence microscope, and the ratio of fluorescence intensity of the cells from excitation at the 340 nm versus 380 nm wavelengths was quantified using Fiji [72].

## $^{25}$Mg uptake assay

HEK-293 cells were seeded in the morning at ±60% confluence and then transfected at the end of the day (5 μg DNA per Petri dish with Lipofectamine 2000 at 1:2 DNA:Lipofectamine 2000 ratio) with mock or CNNM4 constructs. The next day, cells were reseeded on poly-L-lysine (Sigma, St Louis, Missouri, USA) coated 12-well plates and simultaneously induced with 5 μg/mL tetracycline. $Mg^{2+}$ uptake was determined using a stable $^{25}$Mg isotope (Cortecnet, Voisins Le Bretonneux, France), which has a natural abundance of ±10%. Cells were washed with basic uptake buffer (125 mM NaCl, 5 mM KCl, 0.5 mM $CaCl_2$, 0.5 mM $Na_2HPO_4$, 0.5 mM $Na_2SO_4$, 15 mM HEPES/NaOH, pH 7.5) and subsequently placed in basic uptake buffer containing 1 mM $^{25}$Mg (purity ±98%) for 10 minutes. After washing 3 times with ice-cold PBS, the cells were lysed in $HNO_3$ (≥65%, Sigma) and subjected to ICP-MS (inductively coupled plasma mass spectrometry) analysis.

## Electrophysiological recordings

The voltage-clamp technique was used to evaluate the whole-cell currents of TRPM7 expressed in HEK-293 cells as described [51]. Briefly, whole-cell current recordings of TRPM7-expressing cells were elicited by voltage stimuli lasting 250 ms delivered every 1 second using voltage ramps from –100 to +100 mV. Data were digitized at 2 or 5 kHz and digitally filtered offline at 1 kHz. The internal pipette solution for macroscopic current recordings contained (in mM) 145 Cs-methanesulfonate, 8 NaCl, 10 EGTA, and 10 HEPES, pH adjusted to 7.2 with CsOH. The extracellular solution for whole-cell recordings contained (in mM) 140 NaCl, 5 KCl, 2

CaCl$_2$, 10 HEPES, and 10 glucose, pH adjusted to 7.4 (NaOH). For rescue experiments, transient transfection of pCMV-4A-CNNM4 expressed CNNM4 at high levels, which reduced TRPM7 protein expression, made measurement of whole-cell currents too variable for rescue experiments. Placing CNNM4 in vectors with weak promoters still caused too much variability, presumably because of the negative effect of CNNM4 on TRPM7 protein expression. To overcome this problem, pCuO-MCS-CNNM4-FLAG was employed to achieve sustained and modest reexpression of CNNM4 in 293-M7-ΔCNNM3/4 cells, with CNNM4 expression induced upon application of cumate to the cell media. To achieve sustained episomal expression, cells transfected with pCuO-MCS-CNNM4-FLAG and then were selected by application of puromycin to the growth media. Surviving cells were used for the rescue experiments shown in Fig 4A–4D.

### Statistical analysis

Unless otherwise indicated, statistical analysis used 1-way ANOVA plus Student *t* test (2 tailed). A *p*-value of less than 0.05 was considered significant. Error bars indicate standard error.

### Supporting information

**S1 Table. Mass spectrometry reveals TRPM7-interacting proteins.** For identification of TRPM7 interacting proteins by LC-MS/MS, purified TRPM7 was resolved in a Bis-Tris polyacrylamide gel, and each gel band was subjected to in-gel reduction, alkylation, tryptic digestion, and peptide extraction with a standard protocol. Peptides were solubilized in 0.1% trifluoroacetic acid, and analyzed by Nano LC-MS/MS with Dionex Ultimate 3000 RLSC Nano System interfaced with a Velos-LTQ-Orbitrap (Thermo Fisher Scientific) or a QExactive HF (Thermo Fisher Scientific) based on instrument availability. Shown are results from samples immunopurified from LTRPC7 cells in the absence of tetracycline to keep TRPM7 expression levels low and from control cells (293T). The table shows proteins that were identified in 2 experimental replicates of samples from LTRPC7 cells but not in in control 293T cells. (XLSX)

**S1 Raw Images. Unprocessed images of blots from Figs 1, 2 and 4 and S1, S3, S5, S6, S8, S10–S12 and S16 Figs.** (PDF)

**S1 Data. Underlying data for Figs 1–4 and S2 and S6–S12 Figs.** (XLSX)

**S1 Fig. CNNMs interact with TRPM6.** HA-TRPM6 was coexpressed with FLAG-tagged CNNM1-4 in HEK-293T cells, and the channel was immunoprecipitated with HA-agarose. CNNM1 and CNNM3 strongly interacted with TRPM6. An interaction between TRPM6 with CNNM2 and CNNM4 was not observed. CNNMs were not immunoprecipitated in HEK-293T cells N.T. with HA-TRPM6. Unprocessed images of blots are shown in S1 Raw Images. N.T., not transfected. (TIF)

**S2 Fig. Expression of CNNMs suppress expression of EGFP. (A)** EGFP was cotransfected with CNNM1-4 in HEK-293T cells and GFP protein expression was assessed by fluorescence microscopy. Coexpression of CNNM2 and CNNM4 with GFP significantly reduced GFP protein expression, most likely as a result of a decrease in intracellular Mg$^{2+}$. **(B)** 100 cells were randomly selected for quantification. *n* = 100. * indicates a *p*-value of less than 0.05. The

underlying data for this figure can be found in S1 Data.
(TIF)

**S3 Fig. TRPM7 interacts with native CNNMs in different cell lines. (A–D)** HA-TRPM7 was overexpressed in the indicated cells and was the immunoprecipitated with HA-agarose. Native CNNM3 and CNNM4 coimmunoprecipitated with HA-TRPM7 transiently transfected into Hela cells. In OK cells transiently transfected with HA-TRPM7, only native CNNM3 was detected as interacting with HA-TRPM7. In HAP1 cells, both CNNM3 and CNNM4 could be weakly be coimmunoprecipitated with HA-TRPM7, which was expressed by viral transduction using the Ad-TRPM7-HA adenovirus. Note that the lower band in the lysate of the CNNM4 blot is a nonspecific band. In RPTEC/TERT cells transduced with Ad-TRPM7-HA, native CNNM3 coimmunoprecipitated with overexpressed HA-TRPM7. **(E)** Native TRPM7 is highly expressed in ZR-75-1 cells. Immunoprecipitation of native TRPM7 efficiently immunoprecipitated native CNNM4. Unprocessed images of blots are shown in S1 Raw Images. OK, opossum kidney proximal tubule.
(TIF)

**S4 Fig. Localization of TRPM7 and CNNM3 in HEK-293T cells. (A)** Confocal images taken from HEK-293T cells transfected with FLAG-CNNM3. Shown are images of FLAG-CNNM3 and the endogenous Na/K-ATPase, a plasma membrane marker. CNNM3 is often found at the plasma membrane colocalized with the Na/K-ATPase. **(B)** Confocal images taken from HEK-293T cells transfected with HA-TRPM7 and FLAG-CNNM3.TRPM7 can be found colocalized with CNNM3 at the cell border but is also observed intracellularly. Dashed white boxes indicate regions of interest that were enlarged. Scale bar = 20 μM.
(TIF)

**S5 Fig. CNNM3 is vicinal to TRPM7.** Proximity-dependent BioID is a recently developed method that allows the identification of proteins in the close vicinity (10 to 30 nm) of a protein of interest in living cells. We used the optimized *E. coli* BirA biotin ligase (TurboID) to create a fusion protein between CNNM3 and FLAG-tagged TurboID (CNNM3-FLAG-TurboID). As a negative control, we used EYFP fused to FLAG-tagged TurboID (YFP-FLAG-TurboID). CNNM3-TurboID and YFP-TurboID were transfected into 293-TRPM7 cells expressing FLAG-TRPM7. The top blot shows equal expression of TRPM7 in the lysate. In cells expressing TRPM7 with CNNM3-FLAG-TurboID treated with biotin, SA was able to efficiently purify biotinylated TRPM7. By contrast, for cells expressing the TRPM7 with the negative control YFP-TurboID, no biotinylated TRPM7 was purified with the SA. Blots show expression of CNNM3-TurboID and YFP-TurboID in cell lysates and following purification with SA. Unprocessed images of blots are shown in S1 Raw Images. BioID, biotin identification; SA, streptavidin agarose.
(TIF)

**S6 Fig. The TRPM7 channel and its pore is required for CNNM-mediated divalent cation entry in HEK-293 cells. (A)** Zinc influx assay using the Fluo-Zin-3 $Zn^{2+}$ indicator was used to monitor the intracellular concentration of $Zn^{2+}$ in intact cells. Overexpression of CNNM2 (0.15 **μg** CNNM2 DNA) in WT HEK-293T cells 293T(WT) elicited an increase cellular $Zn^{2+}$. By contrast, overexpression of CNNM2 (using 0.6 μg instead of 0.15 μg CNNM2 DNA) in HEK-293T cells deficient in TRPM7 293T(ΔM7) did not increase cellular $Zn^{2+}$. Shown are images taken at a time point 5 to 10 minutes after application of 30 μM $ZnCl_2$. These data indicate that the native TRPM7 channel is required for CNNM-mediated divalent influx in HEK-293 cells. White scale bar = 100 μM. **(B)** Quantification of the results from (A). A total of 100 cells were randomly selected for quantification. $n = 100$. * indicates a $p$-value of less than 0.05.

(C) Western blot showing expression of CNNM2 in the cells used for the experiment described in (A). (D) Separate time course measurements were also acquired to evaluate how CNNM2 overexpression affects the rate of $Zn^{2+}$ influx compared in 293T(WT) versus 293T($\Delta$M7) cells. The TRPM7 channel inhibitor NS8593 (10 **μ**M) was employed to further investigate whether the CNNM2-mediated increase in Fluo-Zin-3 fluorescence upon CNNM2 overexpression in 293T(WT) was dependent on TRPM7 channel function. HBSS media was replaced with HBSS containing 30 μM $ZnCl_2$ for the period indicated. The fluorescence intensity of the cells (mean of 50 cells) were quantified for each time point. (E) Zinc influx assay using the Fluo-Zin-3 $Zn^{2+}$ indicator was used to monitor the intracellular concentration of $Zn^{2+}$ in intact cells. HA-tagged mouse TRPM7 (WT) and TRPM7-E1047K HEK-293 expressing cells 293-M7(E1047K) were used for these experiments (see Methods). Overexpression of CNNM2 with WT TRPM7, but not the TRPM7-E1047K pore-inactive mutant increased $Zn^{2+}$ uptake. However, no $Zn^{2+}$ uptake was observed when a channel-inactive mutant was coexpressed with CNNM2. White scale bar = 100 μM. (F) Quantification of the results from (E). A total of 100 cells were randomly selected for quantification. $n = 100$. * indicates a $p$-value of less than 0.05. (G) Western blot showing expression of TRPM7 and CNNM2 from the experimental samples described in (E). (H) Separate time course measurements were also acquired to demonstrate rate of $Zn^{2+}$ influx in 293, 293(M7), and 293M7(E1047K) cells transfected with CNNM2. HBSS media was replaced with HBSS containing 30 μM $ZnCl_2$ for the period indicated. The fluorescence intensity of the cells (mean of 50 cells) were quantified for each time point. Unprocessed images of blots are shown in S1 Raw Images. The underlying data for this figure can be found in S1 Data. HBSS, Hanks' balanced salt solution; WT, wild-type.
(TIF)

**S7 Fig. The TRPM7 channel is required for CNNM-mediated divalent cation entry in HAP1 cells.** (A) A Zinc influx assay using the Fluo-Zin-3 $Zn^{2+}$ indicator was used to monitor the intracellular concentration of $Zn^{2+}$ in intact cells. Shown are images taken at a time point 5 to 10 minutes after application of 30 μM $ZnCl_2$. Overexpression of CNNM2 in WT HAP1 cells HAP1(WT) using a recombinant adenovirus expressing FLAG-CNNM2 elicited an increase in cellular $Zn^{2+}$. By contrast, overexpression of CNNM2 in HAP1 cells deficient in *TRPM7* HAP1($\Delta$M7) did not significantly increase cellular $Zn^{2+}$. Expression of LacZ was used as a negative control. These data indicate that the native TRPM7 channel is required for CNNM-mediated divalent influx in HAP1 cells. White scale bar = 100 μM. (B) Quantification of the data in (A). A total of 100 cells were randomly selected for quantification. $n = 100$. * indicates a $p$-value of less than 0.05. The underlying data for this figure can be found in S1 Data. WT, wild-type.
(TIF)

**S8 Fig. CNNM4 but not SLC41A1 stimulates TRPM7-mediated divalent cation entry.** (A) A Zinc influx assay using the Fluo-Zin-3 $Zn^{2+}$ indicator was used to monitor the intracellular concentration of $Zn^{2+}$ in intact cells. Shown are images taken at a time point between 5 and 10 minutes after application of 30 μM $ZnCl_2$. In parallel experiments, intracellular $Mg^{2+}$ levels were assessed using the Mag-Fluo-4 $Mg^{2+}$ indicator. White scale bar = 100 μM. (B) Quantification of Mag-Fluo-4 results from (A). A total of 100 cells were randomly selected for quantification. $n = 100$. * indicates a $p$-value of less than 0.05. (C) Separate control experiments were performed using the Mag-Fura-2 dye to show that the magnesium levels of 293-TRPM7 cells expressing CNNM4 as well as SLC41A1 decreased compared to 293-TRPM7 cells alone. Plotted is the ratio of the fluorescence intensity at 510 nm from 340 nm versus 380 nm excitation. 30 cells were randomly selected for quantification. $n = 30$. * indicates a $p$-value of less than 0.05. (D) Quantification of Fluo-Zin-3 $Zn^{2+}$ indicator fluorescence intensity from (A). A total

of 100 cells were randomly selected for quantification. $n = 100$. * indicates a $p$-value of less than 0.05. **(E)** Western blot evaluating expression of CNNM4, SLC41A1, and TRPM7 in the indicated cell lines. **(F)** Confocal images taken from HEK-293T cells transfected with MYC-SLC41A1 and FLAG-CNNM3. SLC41A1 can be found colocalized with CNNM3 at the cell border but is also observed intracellularly, which may be due to the effects of its overexpression. Scale bar = 20 μM. **(G)** Separate time course measurements were also acquired to demonstrate rate of $Zn^{2+}$ influx in the different cell lines. HBSS media was replaced with HBSS containing 30 μM $ZnCl_2$ for the period indicated. The fluorescence intensity of the cells (mean of 50 cells) were quantified for each time point. Unprocessed images of blots are shown in S1 Raw Images. The underlying data for this figure can be found in S1 Data. HBSS, Hanks' balanced salt solution.
(TIF)

**S9 Fig. KO of CNNMs in independent cells lines show similar loss of TRPM7 function. (A)** Zinc influx assay using the Fluo-Zin-3 $Zn^{2+}$ indicator was used to monitor TRPM7 function in independent cell lines in which *CNNM3*, *CNNM4*, and both *CNNM3* and *CNNM4* were deleted by CRISPR/Cas-9 from 293-TRPM7 cells (293-M7-ΔCNNM3, 293-M7-ΔCNNM4, 293-M7-ΔCNNM3/4. Shown are images taken at a time point between 5 and 10 minutes after application of 30 μM $ZnCl_2$. Similar loss of TRPM7 function was observed among the 2 independent lines tested, with deletion of *CNNM3* and *CNNM4* producing the largest loss-of-function. All the cells in the assay were treated with tetracycline to induce TRPM7 expression. White scale bar = 100 **μ**M. **(B)** Quantification of the results from (A). A total of 100 cells were randomly selected for quantification. $n = 100$. * indicates a $p$-value of less than 0.05. The underlying data for this figure can be found in S1 Data. KO, knockout.
(TIF)

**S10 Fig. PRL isoforms stimulate TRPM7-mediated divalent cation entry. (A)** Zinc influx assay using the Fluo-Zin-3 $Zn^{2+}$ indicator was used to monitor the intracellular concentration of $Zn^{2+}$ in intact cells. Shown are images taken at a time point between 5 and 10 minutes after application of 30 μM $ZnCl_2$. Coexpression of the PRL isoforms PRL-1, PRL2, and PRL-3 all stimulated TRPM7-dependent $Zn^{2+}$ uptake. White scale bar = 100 μM. **(B)** Quantification of the data from (A). A total of 100 cells were randomly selected for quantification. $n = 100$. * indicates a $p$-value of less than 0.05. **(C)** Western blot demonstrating expression levels of TRPM7 and PRL isoforms in the experimental samples from (A). **(D)** Separate time course measurements were also acquired to demonstrate rate of $Zn^{2+}$ influx in the different cell lines. HBSS media was replaced with HBSS containing 30 **μ**M $ZnCl_2$ for the period indicated. The fluorescence intensity of the cells (mean of 50 cells) were quantified for each time point. **(E)** Coexpression of WT PRL-2 (PRL-2(WT)) but not the CNNM-binding deficient mutant PRL-2(R017E) stimulates TRPM7-dependent $Zn^{2+}$ uptake. White scale bar = 100 μM. **(F)** Quantification of the data from (E). A total of 100 cells were randomly selected for quantification. $n = 100$. * indicates a $p$-value of less than 0.05. **(G)** Western blot demonstrating expression levels of TRPM7, PRL-2(WT), PRL-2(R107E) in the experimental samples from (E). **(H)** Separate time course measurements were also acquired to demonstrate rate of $Zn^{2+}$ influx in the different cell lines. HBSS media was replaced with HBSS containing 30 μM $ZnCl_2$ for the period indicated. The fluorescence intensity of the cells (mean of 50 cells) were quantified for each time point. Unprocessed images of blots are shown in S1 Raw Images. The underlying data for this figure can be found in S1 Data. HBSS, Hanks' balanced salt solution; PRL, phosphatase of regenerating liver; WT, wild-type.
(TIF)

**S11 Fig. Effect of mutants that disable CNNM Mg²⁺ efflux activity on TRPM7-mediated Zn²⁺ influx.** (**A**) A Zinc influx assay using the Fluo-Zin-3 Zn²⁺ indicator was used to monitor the intracellular concentration of Zn²⁺ in intact cells. Shown are images taken at a time point between 5 and 10 minutes after application of 30 μM ZnCl₂. Coexpression of mutational variants of CNNM2 known to disrupt CNNM2 Mg²⁺ efflux activity also affected TRPM7-mediated increases in intracellular free Zn²⁺ to varying degrees. (**B**) Quantification of the data from (A). A total of 50 cells were randomly selected for quantification. $n = 50$. * indicates a $p$-value of less than 0.05. (**C**) Separate control experiments were performed using the Mag-Fura-2 dye to show that the magnesium levels of 293-TRPM7 cells expressing CNNM2 are decreased compared to 293-TRPM7 cells alone and that CNNM2 mutants used in (A) do not effectively lower intracellular Mg²⁺ levels as effectively as WT CNNM2. Plotted is the ratio of the fluorescence intensity at 510 nm from 340 nm versus 380 nm excitation. A total of 50 cells were randomly selected for quantification. $n = 50$. * indicates a $p$-value of less than 0.05. (**D**) Western blot validating expression of TRPM7 and CNNM2 WT and mutants. Vinculin is shown as a loading control. (**E**) A Zinc influx assay using the Fluo-Zin-3 Zn²⁺ indicator was used to monitor the intracellular concentration of Zn²⁺ in intact cells. Shown are images taken at a time point between 5 and 10 minutes after application of 30 μM ZnCl₂. Coexpression of mutational variants of CNNM4 known to disrupt CNNM4 Mg²⁺ efflux activity also affected TRPM7-mediated increases in intracellular free Zn²⁺ to varying degrees. Strikingly, the CNNM4(S196P) mutant, which is reported to disrupted Mg²⁺ binding to the CNNM4 transmembrane domain, enhanced Zn²⁺ influx compared to WT CNNM4 and the other tested variants. (**F**) Quantification of the data from (E). A total of 50 cells were randomly selected for quantification. $n = 50$. * indicates a $p$-value of less than 0.05. (**G**) Separate control experiments were performed using the Mag-Fura-2 dye to show that the magnesium levels of 293-TRPM7 cells expressing CNNM4 are decreased compared to 293-TRPM7 cells alone and that CNNM4 mutants used in (E) do not effectively lower intracellular Mg²⁺ levels as effectively as WT CNNM4. Plotted is the ratio of the fluorescence intensity at 510 nm from 340 nm versus 380 nm excitation. A total of 50 cells were randomly selected for quantification. $n = 50$. * indicates a $p$-value of less than 0.05. (**H**) To validate the impact of the CNNM4 mutants on Zn²⁺ influx, separate time course measurements were also acquired to demonstrate the rate of Zn²⁺ influx in the different cell lines. HBSS media was replaced with HBSS containing 30 μM ZnCl₂ for the period indicated. The fluorescence intensity of the cells (mean of 50 cells) were quantified for each time point. (**I**) Western blot validating expression of TRPM7 and CNNM4 WT and mutants. Vinculin is shown as a loading control. Unprocessed images of blots are shown in S1 Raw Images. The underlying data for this figure can be found in S1 Data. HBSS, Hanks' balanced salt solution; WT, wild-type.
(TIF)

**S12 Fig. Effect of ARL15 overexpression and knockdown on TRPM7-mediated Zn²⁺ influx.** (**A**) Zinc influx assay using the Fluo-Zin-3 Zn²⁺ indicator was used to monitor the intracellular concentration of Zn²⁺ in intact cells. Shown are images taken at a time point between 5 and 10 minutes after application of 30 μM ZnCl₂. Coexpression of the ARL15 protein suppressed TRPM7-dependent Zn²⁺ uptake compared to nontransfected cells. White scale bar = 100 μM. (**B**) Quantification of the data from (A). A total of 50 cells were randomly selected for quantification. $n = 50$. * indicates a $p$-value of less than 0.05. (**C**) Zinc influx assay using Fluo-Zin-3 Zn²⁺ indicator was used to monitor the intracellular concentration of Zn²⁺ in intact cells. Shown are images taken at a time point between 5 and 10 minutes after application of 30 μM ZnCl₂. Coexpression of siRNAs targeting the ARL15 protein increased TRPM7-dependent Zn²⁺ uptake compared to a control siRNA. White scale bar = 100 μM. (**D**) Quantification of

the data from (A). A total of 50 cells were randomly selected for quantification. $n = 50$. * indicates a $p$-value of less than 0.05. (**E**) Western blot validating ability of 2 independent siRNAs targeting ARL15 to knockdown expression of ARL15 compared to the nonsilencing control. Unprocessed images of blots are shown in S1 Raw Images. The underlying data for this figure can be found in S1 Data.
(TIF)

**S13 Fig. Localization of overexpressed FLAG-CNNM isoforms in OK cells.** Shown are confocal microscopy images taken of the top (apical), middle, and bottom of cells. FLAG-tagged CNNM proteins were stained with the rabbit monoclonal anti-FLAG antibody. NHERF-1, an apical membrane marker that localizes to microvilli on polarized epithelial cells, was stained using a mouse monoclonal antibody. FLAG-CNNM isoforms were transiently transfected in OK cells and stained following a standard protocol (see Methods). Shown are the localization for (**A**) CNNM1, (**B**) CNNM2, (**C**) CNNM3, and (**D**) CNNM4. Scale bar = 20 μM. OK, opossum kidney proximal tubule.
(TIF)

**S14 Fig. Localization of overexpressed FLAG-CNNM isoforms in Caco-2 cells.** Shown are confocal microscopy images taken of the top (apical), middle, and bottom of cells. FLAG-CNNM2 and FLAG-CNNM4 were stained with the rabbit monoclonal anti-FLAG antibody. NHERF-1, an apical membrane marker that localizes to microvilli on polarized epithelial cells, was stained with a mouse monoclonal antibody. FLAG-CNNM2 and FLAG-CNNM4 isoforms were virally transduced into Caco-2 cells and stained following a standard protocol (see Methods). Shown are the localization for (**A**) CNNM2 and (**B**) CNNM4. Scale bar = 20 μM.
(TIF)

**S15 Fig. Localization of overexpressed FLAG-CNNM isoforms in RPTEC/TERT1 cells.** Shown are confocal microscopy images taken of the top (apical), middle, and bottom of cells. FLAG-CNNM2 and FLAG-CNNM4 were stained with the rabbit monoclonal anti-FLAG antibody. NHERF-1, an apical membrane marker that localizes to microvilli on polarized epithelial cells, was stained with a mouse monoclonal antibody. FLAG-CNNM2 and FLAG-CNNM4 isoforms were virally transduced into RPTEC/TERT1 cells and stained following a standard immunofluorescence protocol (see Methods). Shown are the localization for (**A**) CNNM2 and (**B**) CNNM4. Scale bar = 20 μM.
(TIF)

**S16 Fig. CNNM isoforms form specific heteroligomers.** (**A**) FLAG-CNNM1 and FLAG-CNNM2 were individually expressed in HEK-293T cells. Immunoprecipitation of FLAG-CNNM1 effectively immunopurified native CNNM4, whereas native CNNM3 was immunopurified to a lesser degree. Immunoprecipitation of FLAG-CNNM2 immunopurified native CNNM3 but not native CNNM4. (**B**) FLAG-CNNM3 and FLAG-CNNM4 were individually expressed in HEK-293T cells. Immunoprecipitation of FLAG-CNNM4 efficiently immunopurified native CNNM1, whereas immunoprecipitation of FLAG-CNNM3 did not immunopurify native CNNM1. (**C**) FLAG-CNNM3 was expressed in HEK-293T cells. Immunoprecipitation of FLAG-CNNM3 was able to coimmunopurify native CNNM4. (**D**) FLAG-CNNM4 was expressed in HEK-293T cells. Immunoprecipitation of FLAG-CNNM4 was able to coimmunopurify native CNNM3. (**E**) Immunoprecipitation of native CNNM3 is able to coimmunopurify native CNNM4. Anti-myc antibody was used as a negative control for immunoprecipitation. Indicated is the nonspecific band derived from the antibody used for the immunoprecipitation. Below this band is CNNM4, as indicted by the arrow. (**F**) Immunoprecipitation of native CNNM4 is able to coimmunopurify native CNNM3. The anti-myc

antibody was used as a negative control for immunoprecipitation. Indicated is the nonspecific band derived from the antibody used for the immunoprecipitation. Below the nonspecific band is CNNM3, as indicted by the arrow. Unprocessed images of blots are shown in S1 Raw Images.
(TIF)

## Acknowledgments

We thank Dr. Hiroaki Miki (Osaka University, Japan), Dr. Andrew Scharenberg (University of Washington), Dr. Vladimir Chubanov (University of Munich), and Dr. David Clapham (Janielia Research Campus, HHMI) for sharing their reagents.

## Author Contributions

**Conceptualization:** Zhiyong Bai, Lixia Yue, Loren W. Runnels.

**Investigation:** Zhiyong Bai, Jianlin Feng, Gijs A. C. Franken, Namariq Al'Saadi, Na Cai, Albert S. Yu, Liping Lou, Yuko Komiya, Joost G. J. Hoenderop, Jeroen H. F. de Baaij, Lixia Yue.

**Project administration:** Loren W. Runnels.

**Supervision:** Loren W. Runnels.

**Writing – original draft:** Zhiyong Bai.

**Writing – review & editing:** Zhiyong Bai, Joost G. J. Hoenderop, Jeroen H. F. de Baaij, Lixia Yue, Loren W. Runnels.

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
