## [Editor Report · Decision Letter 0]

29 Jun 2020

Dear Dr Runnels, 

Thank you for submitting your manuscript entitled "CNNM Proteins Regulate the TRPM7 Channel to Control Cellular Divalent Cation Entry" for consideration as a Short Report by PLOS Biology.

Your manuscript has now been evaluated by the PLOS Biology editorial staff as well as by an academic editor with relevant expertise and I am writing to let you know that we would like to send your submission out for external peer review.

Please re-submit your manuscript within two working days, i.e. by Jul 01 2020 11:59PM.

Kind regards,

Ines

--

Ines Alvarez-Garcia, PhD

Senior Editor

PLOS Biology

Carlyle House, Carlyle Road

Cambridge, CB4 3DN

+44 1223–442810

---

## [Decision Letter · Decision Letter 1]

13 Aug 2020

Dear Dr Runnels,

Thank you very much for submitting your manuscript "CNNM Proteins Regulate the TRPM7 Channel to Control Cellular Divalent Cation Entry" for consideration as a Short Report at PLOS Biology. Your manuscript has been evaluated by the PLOS Biology editors, by an Academic Editor with relevant expertise, and by three independent reviewers.

The reviews of your manuscript are appended below. You will see that the reviewers find the work potentially interesting. However, based on their specific comments and following discussion with the Academic Editor, I regret that we cannot accept the current version of the manuscript for publication. We remain interested in your study and we would be willing to consider resubmission of a comprehensively revised version that thoroughly addresses all the reviewers' comments. We cannot make any decision about publication until we have seen the revised manuscript and your response to the reviewers' comments. Your revised manuscript would be sent for further evaluation by the reviewers.

***IMPORTANT: 

Having discussed the reviews with the Academic Editor, we think you should add more data to address the issues raised by the reviewers. A mere response/rebuttal to the criticism will be insufficient for eventual publication. This is particularly important in light of the comments made by reviewers 1 and 3. We think, in agreement with the Academic Editor, that the quantitative interpretation of ion concentration is substantial as it requires reevaluation and repetition of numerous experiments. Another important concern is related to the claim that both proteins would reside on opposite sides of polarized epithelia and thus not come into contact under physiological conditions.

We appreciate that these requests represent a great deal of extra work, and we are willing to relax our standard revision time to allow you six months to revise your manuscript. We expect to receive your revised manuscript within 6 months. Depending on the amount of new data you add during revision, we will reconsider whether your manuscript should be handled as a Short Report or as a Full Research Article.

If you find it useful, we would be happy to discuss with the Academic Editor a revision proposal that includes your planned experiments.”

**IMPORTANT - SUBMITTING YOUR REVISION**

*Resubmission Checklist*

*Published Peer Review*

*PLOS Data Policy*

*Blot and Gel Data Policy*

Sincerely,

Gabriel Gasque

Senior Editor

in behalf of

Ines Alvarez-Garcia, PhD,

Senior Editor,

ialvarez-garcia@plos.org,

PLOS Biology

REVIEWS:

Reviewer #1: In this manuscript, Bai et al. report a novel role of CNNM proteins in regulating TRPM7 cation channel. CNNM proteins are involved in Mg transport, but there has been a controversy over its precise function. The authors found that CNNM proteins bind to TRPM7 and stimulates its channel activity. TRPM7 is permeable to Mg and has been shown to play an important role in cellular and organismal Mg homeostasis. The authors claim that this study has revealed a crucial importance of CNNM proteins in regulating intracellular Mg levels by coordinating its direct role in Mg efflux and indirect role in Mg influx via TRPM7. However, the presented data are preliminary and not solid enough to support the authors' claim. In addition, there are many errors in the text. Substantial modifications and additional experiments will be needed to improve the quality of this manuscript to the level appropriate for publication in PLOS Biology. Major and minor problems are listed as follows.

Major problems:

1: The authors claim that CNNM proteins have dual functions in both Mg efflux by themselves and Mg influx by stimulating TRPM7. The idea itself sounds very attractive because it can reconcile the controversy over the function of CNNM proteins. However, the presented data clearly show that CNNM expression decreases Mg levels irrespective of the TRPM7 expression status (Fig 3C). Therefore, Mg efflux seems to be the dominant function of CNNM proteins, even if Mg influx may also be stimulated. The authors need to show that CNNM actually stimulates Mg influx in the presence of TRPM7 to validate their claim. It should also be noted that there are serious problems in Mg imaging analyses, which were used to quantify Mg levels (see the following comment no. 2).

 Related to this issue, CNNM proteins are known to be associated with various congenital diseases, and several disease-related missense mutants have been shown to abolish Mg efflux activity. Using these mutant forms of CNNM proteins, the authors may be able to discriminate the activity for Mg influx via TRPM7 from that for Mg efflux. Experiments using such CNNM mutants are desirable.

2: There are serious problems in imaging analyses for Zn and Mg. As for Zn imaging, the authors show only a single fluorescence image, which was obtained at 5-10 min after the 30 microM Zn supplementation, for each sample. However, it cannot be excluded that the results just reflect uneven incorporation of the fluorescent probe. In addition, time-lapse analyses before and after the Zn addition to the medium should be performed to quantify Zn influx into cells.

 Mg imaging experiments were done using Mag-Fluo4. It may be a good probe for time-lapse monitoring of the dynamic change in Mg levels in the same cell, but it is not ratiometric. Therefore, it cannot be used for quantifying absolute Mg levels, to compare different samples. For this purpose, the authors should use ratiometric probes for Mg with appropriate calibrations. Moreover, it should be noted that these Mg probes react not only with Mg but also strongly with Ca. TRPM7 channel is permeable to Ca, and thus, the results obtained by using fluorescent Mg probes need to be carefully interpreted. The authors should verify that the fluorescence signals in Mg imaging experiments properly reflect Mg levels, but not Ca levels, when analyzing TRPM7 functions.

3: CNNM proteins are mainly expressed in polarized epithelial cells in vivo and localized at their basolateral membrane, whereas TRPM proteins are localized at the apical membrane. Therefore, they are secluded from each other in epithelial cells. It is uncertain whether the authors' conclusion based on the analyses using non-polarized culture cells can be generalized. The authors should perform experiments using epithelial cells and investigate the localization, complex formation, and functional relationship of CNNM and TRPM. 

4: The authors show the data of TRPM7-CNNM complex formation using culture cells ectopically expressing TRPM7. They should investigate whether endogenous TRPM7 proteins associate with endogenous CNNM proteins. Also, it should be addressed whether the complex formation status is affected by PRL expression or Mg availability, such as Mg depletion from the medium and Mg supplementation.

Minor problems:

Fig 1B: The estimated size of the bands in the IP samples seems to be larger than that in the lysate in the same gel.

Fig 1C: The authors claim that the expressed proteins co-localize at the cell border. However, the cells have become rounded and the most part is occupied by the nuclei. Therefore, it is difficult to determine the precise localization of these proteins based on the immunofluorescent images shown here. Ectopically expressed membrane proteins often aggregate intracellularly, and thus, more careful investigations, such as co-staining with plasma membrane markers with appropriate controls and quantifications, are needed.

Fig 1E: The information of the anti-TRPM7 antibody should be provided in the methods section. 

Fig 2C: The CNNM4-immunoblotting result shows irregular bands in CNNM3&4-KO cells and CNNM3-KO cells. What do these signals represent? Furthermore, the authors performed immunoblotting analyses to verify these KO cells, but the changes at the genome level should be directly confirmed by DNA sequencing around the CRISPR-Cas9 target site and shown in the figure.

Fig 2E: The result is not clear enough to justify the authors' claim. In addition, no data for validating the specific labeling of membrane proteins are presented.

Fig 2F-H: Experiments using PRL-2 overexpression need appropriate controls. The results of R107E mutant shown in S9 fig should be moved here.

Fig 2G: The information of the anti-PRL-2 antibody should be provided in the methods section.

Fig 3A: The authors should perform appropriate statistical analyses to validate the claim.

S1 Table: Many candidate proteins are identified in MS analyses. The authors should explain the reason why they chose CNNM proteins for further analyses. Also, experimental details of MS analyses should also be described.

S7 fig: Immunoblotting and immunofluorescence data for SLC41A1 should be shown in the figure to verify its proper expression and localization at the plasma membrane.

Other minor errors:

Line 25: Remove "it".

Line 133: D and E should be E and F, respectively.

Line 147: Fig 1D should be Fig 1E.

Line 148: overexpression "of" either

Line 168: S7 fig 6 should be S7 fig B.

Line 174: Fig 1C,E should be Fig 1D,F.

Line 179: Remove "Fig 2A,B".

Line 244: "evaluate"

Line 251: Fig 3C should also be refferred here.

Line 283: The voltage information should be indicated.

Line 389: "were" resolved

Line 443: Experimental details of the ICP-MS analyses should be described.

Line 454: transfection "of"

Line 463: ANOVA should be used for statistical analyses. Also, two-tailed test would be appropriate.

Line 674: "LacZ" in the figure should be mentioned in the legend.

Reviewer #2: The manuscript provides convincing data supporting the hypothesis that CNNM proteins are regulatory/accessory proteins enhancing divalent ion flux through the channel domain of the TRPM7 bifunctional protein. In the absence of TRPM7, CNNMs carry an important magnesium efflux role.

While the data are comprehensive and logically presented, major confusion remains in regards to the TRPM7 species, antibody tag and expression system used. For example, it seems that three different overexpression systems were used for TRPM7 alone, and all in HEK293 cells: FLAG tag (identified as mouse TRPM7, tet-induction), HA tag (species not identified, also not described in methods as of source etc. Human?), and a third unspecified "stably overexpressing TRPM7" cell line, also not described in the methods (line 141 and beyond). What was the tag, source and overexpression system (tetracycline? Other?) of the "stably overexpressing TRPM7" system? In most cases, TRPM7 overexpression leads to cell detachment and cell loss. How did the authors counteract that in this cell line? The above has to be clarified in order to substantiate the conclusions drawn. 

The single point mutation in the TRPM7 channel domain is cited from previous work by the authors, however, it would be helpful to again mention the source and species and expression system in the methods.

S5B (described line 151 and beyond) clearly shows a doubling in Zn influx in the absence of TRPM7 and overexpression of CNNM2 compared to WT alone. Is this statistically significant? This is in contrast to what is seen in HAP1 cells (S6). The text guides the reader away from this observation and instead provides a lump sum statement that endogenous TRPM7 is required for Zn influx. This might be the case for the HAP1 cells, but in HEK293 the picture might be more differentiated. Perhaps worth briefly discussing (if statistical significance is observed in HEK293)?

The primer sequences for the Turbo ID experiments need to be provided in the Supplemental Data (line 352).

Reviewer #3: In the present paper, Runnels and colleagues identify and characterize CNNM proteins as modulators of TRPM7 function and propose the concept that CNNMs may serve as regulatory subunits of TRPM7 channel complexes in the plasma membrane to fine-tune cellular magnesium entry via TRPM7.

In principle, these are very interesting observations. However, there are a number of critical issues that require further attention.

Conceptual shortcomings:

(1) The physiological relevance of the results is rather limited. In particular, the study mostly relies on proteomic data and functional experiments conducted with cultured cell lines (HEK293) and - with few exceptions - overexpressed proteins. Hence, it remains unknown whether TRPM7 interacts with CNNM and RPL proteins in native tissues.

(2) Mechanistically, the regulatory role of CNNM proteins for TRPM7 activity has been clearly defined. Several important questions remain to be addressed. Thus, it remains elusive which channel characteristics of TRPM7 are affected by CNNM or RPL proteins (open probability, kinetics of activation/rundown, sensitivity to PIP2, Mg2+ and ATP etc.). Does TRPM7 interact with CNNMs transiently or are stable complexes formed? What is the stoichiometry of such heteromeric complexes? Which domains of TRPM7 and CNNM proteins are required for the interaction to occur?

(3) The dominating theme of the Abstract, Introduction and Discussion sections is cellular Mg2+ homeostasis and possible implications of CNNM/RPL proteins in Mg2+ transport. In this regard, there is a kind of disconnect with the actual experiments performed mostly concentrating on biochemical verification of proteomic data, imaging of free Zn2+ levels and patch-clamp assessment of outward monovalent cation currents of TRPM7. As the TRPM7 channel plays a central physiological role in the transport of Zn2+ and Ca2+ (in vitro and in vivo), it appears to be somewhat premature and unjustified to claim that the experiments shown are mainly relevant for Mg2+ homeostasis. 

(4) Nearly all the data shown in the manuscript indicate that CNNM/RPL proteins are not essential (required) TRPM7 function per se. For instance, Fig. 5 clearly shows that TRPM7 currents were only reduced in the absence of CNNM3 and CNNM4. Therefore, the authors should avoid claims that CNNN proteins are 'required' for or 'control' the TRPM7 channel. It would be more appropriate to use terms like 'regulate' or 'modulate'.

(5) The author correctly point out that most of the intracellular Mg2+ is bound to ATP and other metabolites and that free Mg2+ accounts for only ~10% of total Mg2+ levels. It remains unclear how to correlate the reported alterations in Mag-Fluor4 fluorescence and 25Mg2+ uptake with particular changes either in free Mg2+ levels or in total Mg2+ content of the cell. The authors regularly claim in the text that CNNM/RPL are central for cellular Mg2+ homeostasis. To this end, the authors have to provide direct evidence supporting such a conclusion.

(6) The most prominent phenotype of TRPM7-deficient cells is reduced total Mg2+ levels entailing a Mg2+-dependent proliferation block. The authors created CNNM3 and CNNM4 gene-deficient clones which are important and valuable reagents to demonstrate the physiological importance of these proteins. If CNNM proteins are indeed required for TRPM7-mediated Mg2+ uptake, CNNM-deficient cells should somehow mirror the TRPM7 null phenotype. However, the mutant cells appear to grow normally under standard cell culture conditions (the authors did not explicitly state that extra Mg2+ has to be added to the culture medium). These results have to be transparently reported and discussed accordingly.

Methodological shortcomings:

(1) TRPM7 proteome.

Proteomic procedures are not described at all. As gleaned from the short text in the footnote of Table S1, it appears that FLAG-tagged TRPM7 was pulled down using FLAG-specific antibodies in HEK293 cells overexpressing the recombinant protein. The authors have to provide a detailed and fully transparent protocol for all steps of the experiment, including cell lysis conditions, immunoprecipitation, and assessment of the yield and purity of enriched TRPM7 proteins. What was the negative control to rule out unspecific hits? How exactly were raw MS data analysed? What were the biological and technical replicates of these experiments and how as statistical analysis of the data done? How were proteomic data sets assessed qualitatively and quantitatively and what exactly were the criteria and the statistical procedure to identify hits? The authors should explain what is exactly depicted in the '293-TRPM7 peptide count' and '293- peptide count' columns of Table S1. Which proteins are considered as interaction partners of TRPM7? Most proteins are abundantly present in both columns. Are they considered as false-positive hits? What was the ratio of positive and unspecific hits? CNNM3 and CNNM4 proteins are represented by a relatively low frequency of peptides detected. How does this finding support the hypothesis that that CNNMs are relevant interaction partners of TRPM7? Finally, fully annotated proteome results have to be submitted to a publicly accessable database available for reviewers and the scientific community.

(2) IP-based assessment of TRPM7-CNNM interaction.

Many methodological issues are not described. The authors should report all procedures and reagents used for IP experiments. Also, please explain what kind of negative control is shown in Fig. 1A? Presumably, HA-TRPM7 protein was not expressed in HEK293 cells, but the corresponding figure legend does not clearly explain this point. As CNNM proteins are transmembrane proteins, incomplete solubilisation of the membrane fraction is a common limitation of IP approaches in general, i.e. pull-down of TRPM7-containing vesicles together with CNNM proteins may occur. How was the quality of the membrane solubilisation assessed for the experiments shown in Fig. 1?

(3) 'Zinc influx' assay.

In the manuscript Zn2+ imaging experiments reflect the main approach to identify the functional role of CNNM proteins. However, this is the weakest aspect of the manuscript. In many figures (Fig. 1D, 2D, 3D, 4C, S3, S5-S9), the authors used whole-frame images captured at only one time point to measure intracellular Fluo-Zin-3 fluorescence and described this approach as a 'zinc influx' assay. At best, such images can rather crudely report on the steady-state level of free cytosolic Zn2+, but cannot be interpreted as dynamic 'zinc- influx'. Thus, transient expression of CNNM proteins may affect efflux of cytosolic Zn2+ or release of Zn2+ from intracellular compartments (free Zn2+ accounts for only ~1% of the total cellular Zn2+ content). In addition, overexpression of TRPM7 or CNNMs may influence the loading efficiency of Fluo-Zin-3 rather than free Zn2+ levels. Furthermore, the authors randomly picked up cells to quantify the Fluo-Zin-3 fluorescence, but it is unlikely that CNNM cDNAs were transiently transcribed in 100% of cells. How would Fluo-Zin-3 fluorescence look like in un-transfected cells on the same dish? The conventional approach to conduct such dynamic influx studies is based on single-cell imaging over time to also work out the kinetics of changes of Fluo-Zin-3 fluorescence in transfected cells. Also, calibration of Zin-3 fluorescence against reference concentrations of Zn2+ is common practice to verify the physiological impact of alterations of Zin-3 fluorescence and to exclude numerous potential artefacts like movement of cells, bleaching of the probe, and reduced loading of the probe. TRPM7 blockers should be used to demonstrate that changes of Zn2+ uptake are indeed due to increased TRPM7 channel activity. Hence, the author's claim that CNNMs regulate TRPM7-mediated Zn2+ influx is not convincingly supported by the data shown.

(4) Mag-Fluo4 imaging.

As explained above for experiments with Fluo-Zin-3, single snapshots of cells loaded with Mag-Fluor4 do not provide meaningful mechanistic information about the functional role of CNNMs.

---

## [Decision Letter · Decision Letter 2]

10 Aug 2021

Dear Dr Runnels,

Thank you very much for submitting a revised version of your manuscript entitled "CNNM Proteins Regulate the TRPM7 Channel to Control Cellular Divalent Cation Entry" for consideration as a Short Report at PLOS Biology. This revised version of your manuscript has been evaluated by the PLOS Biology editors, the Academic Editor and two of the original reviewers.

In light of the reviews (attached below), we will not be able to accept the current version of the manuscript, but we would welcome re-submission of a revised version that takes into account the reviewers' comments. We cannot make any decision about publication until we have seen the revised manuscript and your response to the reviewers' comments. Your revised manuscript is also likely to be sent for further evaluation by the reviewers.

We expect to receive your revised manuscript within 3 months. 

**IMPORTANT - SUBMITTING YOUR REVISION**

Your revisions should address the specific points made by each reviewer. Having discussed these comments with the Academic Editor, we think that the points raised by reviewer 1 are generally reasonable and that you should address them the best you can, within a reasonable timeframe. If the inclusion of the other suggested CNNM mutants and of the extended localization data would be too time consuming, we would not press for those experiments. In the latter case, you should acknowledge the possibility of overexpression artefacts. Regarding the comments of reviewer 2, we think you should try to integrate the inconclusive results of the CNNM mutants in your manuscript.

Please submit the following files along with your revised manuscript:

*Re-submission Checklist*

*Published Peer Review*

*PLOS Data Policy*

*Blot and Gel Data Policy*

Sincerely,

Gabriel Gasque on behalf of

--

Ines Alvarez-Garcia, PhD

Senior Editor

PLOS Biology

Reviewers’ comments

Rev. 1:

The authors have made a significant effort in this revision and responded to many of the comments on the previous version. However, there still remain several problems that have not been sufficiently addressed or left uncorrected, which are explained below.

1: The experiments using 25Mg (Fig. 3A) clearly show that Mg2+ influx is stimulated by CNNM proteins. However, the critical point of this paper is whether it is ascribed to the "stimulation of TRPM7 activity by the protein-protein interaction with CNNM". The influx of 25Mg may just reflect the TRPM7 function independent of the interaction with CNNM proteins, for maintaining intracellular Mg2+ levels. The Na+ depletion experiments (Fig. 3E) show the increase in Mg2+ levels, even when Mg2+ efflux by CNNM proteins was probably suppressed. However, such Mg2+ increase can be caused by the reverse action of the ectopically expressed CNNM4 proteins under very abnormal condition. CNNM4 normally extrudes Mg2+ in exchange for Na+ influx, but in the absence of extracellular Na+, it can extrude Na+ in exchange for Mg2+ influx in the reverse action. In general, cells have approximately 10 mM intracellular Na+, of which efflux is theoretically sufficient for a few mM level increase in intracellular Mg2+. The authors' data show that Mg2+ increase by Na+ depletion was suppressed by NS8593 (TRPM7 inhibitor) and TRPM7 knockout, but I am wondering how these treatments could suppress the reverse action of the expressed CNNM4 proteins.

Related to this problem, I suggested the authors to use congenital disease-related CNNM mutants in the previous review comment. The authors actually tested the effect of one disease-related point mutant (CNNM2 T568I) and several deletion mutants. However, domain level deletions would certainly affect the protein structure, and the T568I mutation was reported to abolish the interaction with its ligand ATP (J. Biol. Chem. 289, 14731-14739, 2014), which affects the whole structure of the CNNM proteins (Sci. Adv. 7, eabe6140, 2021; Nat. Commun. 12, 4028, 2021). Therefore, these mutants may not be appropriate for discriminating Mg2+ influx function (via TRPM7) from Mg2+ efflux function (by itself). It should be noted here that there are several other point mutations in CNNM4, which are responsible for Jalili syndrome and abolish the Mg2+ efflux function, such as S196P and S200Y, … (Sci. Adv. 7, eabe6140, 2021). In addition, recently reported structural data of its bacterial ortholog revealed that the residues are directly involved in the recognition of Mg2+ in the pore region formed in the membrane-spanning domain (Sci. Adv. 7, eabe6140, 2021). Mutations in these residues are expected to specifically affect the Mg2+ transporting function without so deleterious effects on the whole protein structure. In addition, the possibility of the reverse action of CNNM4 can be excluded using these mutants. The concept of Mg2+ influx function independent of Mg2+ efflux function is the core of this study, and thus, the authors should make the point clear by testing the effect of these congenital disease-related point mutants and showing the results.

2: The authors performed Mg2+ imaging analyses using Magfura2, but showed only the ratio values in the figure. As stated in the previous review comment, the authors should also perform appropriate calibration and show the calculated intracellular Mg2+ concentration, which is needed for proper evaluation of the data.

3: The newly added data of the localization of CNNM proteins (S11-S14) are not convincing. The authors use NHERF1 as an apical marker, but the signals do not show apical localization pattern in the presented data. In addition, localization of membrane proteins in epithelial cells are normally evaluated by vertical section images reconstituted from the stacks of confocal section images. Beautiful examples of apical localization of NHERF1 can be seen in the published papers (Mol. Membr. Biol. 18, 3-11, 2001; Am. J. Physiol. Renal Physiol. 311, F343-F351, 2016), in which the signals are observed only at the top side membrane in a large dot-like manner. In general, cell polarization requires careful handling of cell culture, and I suppose the cells did not polarize well enough for evaluating apical-basolateral localization. Normal localization of marker proteins is prerequisite for evaluating CNNM localization.

However, reading the authors' reply, I understand that further investigation in epithelial cells is not needed in this study. In that case, these preliminary data of expressed CNNM proteins and their descriptions in Discussion should be removed.

4: Ectopically expressed membrane proteins often go to unusual locations. Therefore, localization data of SLC41A1 would be essential for appropriate evaluation of the results, as requested in the previous review comment.

5: In this revision, the authors added sequence information of the CNNM knockout cell lines used in this study (lines 540-547). Several frame-shift deletions in CNNM3 are described, but no information of CNNM4 is provided.

6: In the previous review comment, I listed several minor problems as "other minor points" at the last part. The authors answered properly to most of them, but some are left uncorrected. The following is the list of my comments on the original version and the authors' reply in the rebuttal letter, which are actually not corrected despite the authors' reply. The authors used ANOVA only for analyzing the results of Fig. 3A in this revision, but there are other data that need ANOVA (Figs. 1D, 2B, 2H, 3C, 3D, 4C, 4E, S2B, S8B, S8C, S8D, S9B, S10B, and S10F).

Line 25: Remove "it". - removed

Line 244: "evaluate" - This text is now "evaluated"

Line 389: "were" resolved - "were" was added.

Line 463: ANOVA should be used for statistical analyses. Also, two-tailed test would be appropriate. We have updated the statistical tests used in our analysis.

Rev. 2:

The authors' have addressed my concerns. However, I strongly suggest to include their new negative data on the CNNM mutants (point 1 under Reviewer 1) with a short explanation why the experiments were done and a short discussion. While these data are negative, I feel it is important for the scientific community to know about negative results within the framework of a larger study such as this. It will provide context and likely spark further studies.

---

## [Editor Report · Decision Letter 3]

12 Nov 2021

Dear Dr Runnels,

Thank you for submitting your revised Short Report entitled "CNNM Proteins Regulate the TRPM7 Channel to Control Cellular Divalent Cation Entry" for publication in PLOS Biology. I have now discussed the revision with the rest of the team and obtained advice from the Academic Editor.

Based on the reviews, we will probably accept this manuscript for publication, provided you satisfactorily address the remaining policy-related requests (see below).

In addition, we would like you to consider a suggestion to improve the title:

"CNNM proteins selectively bind to the TRPM7 channel to stimulate divalent cation entry into cells"

We expect to receive your revised manuscript within two weeks. 

*Published Peer Review History*

*Early Version*

Sincerely,

Ines

--

Ines Alvarez-Garcia, PhD,

Senior Editor,

ialvarez-garcia@plos.org,

PLOS Biology

DATA POLICY:

Thank you very much for submitting the data file containing the data underlying all the graphs shown in the figures. We have found several mistakes in the labels that should be amended:

Fig. S6: First Fig. S6E should be labelled S6F, and second Fig. S6E should be S6H

Fig. S8: Second Fig. S8B should be S8C, Fig. S8C should be S8D and Fig. S8F should be S8G - also please make sure you include the labels in this graph.

Fig. S11: Fig. S11E should be S11F, Fig. S11F should be S11G and Fig. S11G should be S11H

Please also make sure that the files deposited in the ProteomeXchange database (PXD026635) are made publicly available or we won't be able to proceed with production.

---

## [Editor Report · Decision Letter 4]

26 Nov 2021

Dear Dr Runnels,

On behalf of my colleagues and the Academic Editor, Raimund Dutzler, I am pleased to say that we can in principle accept your Short Report entitled "CNNM proteins selectively bind to the TRPM7 channel to stimulate divalent cation entry into cells" for publication in PLOS Biology, provided you address any remaining formatting and reporting issues. These will be detailed in an email that will follow this letter and that you will usually receive within 2-3 business days, during which time no action is required from you. Please note that we will not be able to formally accept your manuscript and schedule it for publication until you have any requested changes.

PRESS

Sincerely, 

Ines

--

Ines Alvarez-Garcia, PhD 

Senior Editor 

PLOS Biology
